# Greedy-GQ with Variance Reduction: Finite-time Analysis and Improved Complexity

**Shaocong Ma, Ziyi Chen & Yi Zhou**
Department of ECE
University of Utah
Salt Lake City, UT 84112
{s.ma,u1276972,yi.zhou}@utah.edu

**Shaofeng Zou**
Department of EE
University at Buffalo
Buffalo, NY 14260
szou3@buffalo.edu

## Abstract

Greedy-GQ is a value-based reinforcement learning (RL) algorithm for optimal control. Recently, the finite-time analysis of Greedy-GQ has been developed under linear function approximation and Markovian sampling, and the algorithm is shown to achieve an $\epsilon$-stationary point with a sample complexity in the order of $\mathcal{O}(\epsilon^{-3})$. Such a high sample complexity is due to the large variance induced by the Markovian samples. In this paper, we propose a variance-reduced Greedy-GQ (VR-Greedy-GQ) algorithm for off-policy optimal control. In particular, the algorithm applies the SVRG-based variance reduction scheme to reduce the stochastic variance of the two time-scale updates. We study the finite-time convergence of VR-Greedy-GQ under linear function approximation and Markovian sampling and show that the algorithm achieves a much smaller bias and variance error than the original Greedy-GQ. In particular, we prove that VR-Greedy-GQ achieves an improved sample complexity that is in the order of $\mathcal{O}(\epsilon^{-2})$. We further compare the performance of VR-Greedy-GQ with that of Greedy-GQ in various RL experiments to corroborate our theoretical findings.

## 1 Introduction

In reinforcement learning (RL), an agent interacts with a stochastic environment following a certain policy and receives some reward, and it aims to learn an optimal policy that yields the maximum accumulated reward Sutton & Barto (2018). In particular, many RL algorithms have been developed to learn the optimal control policy, and they have been widely applied to various practical applications such as finance, robotics, computer games and recommendation systems Mnih et al. (2015; 2016); Silver et al. (2016); Kober et al. (2013).

Conventional RL algorithms such as Q-learning Watkins & Dayan (1992) and SARSA Rummery & Niranjan (1994) have been well studied and their convergence is guaranteed in the tabular setting. However, it is known that these algorithms may diverge in the popular off-policy setting under linear function approximation Baird (1995); Gordon (1996). To address this issue, the two time-scale Greedy-GQ algorithm was developed in Maei et al. (2010) for learning the optimal policy. This algorithm extends the efficient gradient temporal difference (GTD) algorithms for policy evaluation Sutton et al. (2009b) to policy optimization. In particular, the asymptotic convergence of Greedy-GQ to a stationary point has been established in Maei et al. (2010). More recently, Wang & Zou (2020) studied the finite-time convergence of Greedy-GQ under linear function approximation and Markovian sampling, and it is shown that the algorithm achieves an $\epsilon$-stationary point of the objective function with a sample complexity in the order of $\mathcal{O}(\epsilon^{-3})$. Such an undesirable high sample complexity is caused by the large variance induced by the Markovian samples queried from the dynamic environment. Therefore, we want to ask the following question.

- *Q1: Can we develop a variance reduction scheme for the two time-scale Greedy-GQ algorithm?*

In fact, in the existing literature, many recent work proposed to apply the variance reduction techniques developed in the stochastic optimization literature to reduce the variance of various TD learning algorithms for policy evaluation, e.g., Du et al. (2017); Peng et al. (2019); Korda & La (2015); Xu et al. (2020). Some other work applied variance reduction techniques to Q-learning algorithms, e.g., Wainwright (2019); Jia et al. (2020). Hence, it is much desired to develop a variance-reduced Greedy-GQ algorithm for optimal control. In particular, as many of the existing variance-reduced RL algorithms have been shown to achieve an improved sample complexity under variance reduction, it is natural to ask the following fundamental question.

- *Q2: Can variance-reduced Greedy-GQ achieve an improved sample complexity under Markovian sampling?*

In this paper, we provide affirmative answers to these fundamental questions. Specifically, we develop a two time-scale variance reduction scheme for the Greedy-GQ algorithm by leveraging the SVRG scheme Johnson & Zhang (2013). Moreover, under linear function approximation and Markovian sampling, we prove that the proposed variance-reduced Greedy-GQ algorithm achieves an $\epsilon$-stationary point with an improved sample complexity $\mathcal{O}(\epsilon^{-2})$. We summarize our technical contributions as follows.

## 1.1 OUR CONTRIBUTIONS

We develop a variance-reduced Greedy-GQ (VR-Greedy-GQ) algorithm for optimal control in reinforcement learning. Specifically, the algorithm leverages the SVRG variance reduction scheme Johnson & Zhang (2013) to construct variance-reduced stochastic updates for updating the parameters in both time-scales.

We study the finite-time convergence of VR-Greedy-GQ under linear function approximation and Markovian sampling in the off-policy setting. Specifically, we show that VR-Greedy-GQ achieves an $\epsilon$-stationary point of the objective function $J$ (i.e., $\|\nabla J(\theta)\|^2 \leq \epsilon$) with a sample complexity in the order of $\mathcal{O}(\epsilon^{-2})$. Such a complexity result improves that of the original Greedy-GQ by a significant factor of $\mathcal{O}(\epsilon^{-1})$ Wang & Zou (2020). In particular, our analysis shows that the bias error caused by the Markovian sampling and the variance error of the stochastic updates are in the order of $\mathcal{O}(M^{-1}), \mathcal{O}(\eta_\theta M^{-1})$, respectively, where $\eta_\theta$ is the learning rate and $M$ corresponds to the batch size of the SVRG reference batch update. This shows that the proposed variance reduction scheme can significantly reduce the bias and variance errors of the original Greedy-GQ update (by a factor of $M$) and lead to an improved overall sample complexity.

The analysis logic of VR-Greedy-GQ partly follows that of the conventional SVRG, but requires substantial new technical developments. Specifically, we must address the following challenges. First, VR-Greedy-GQ involves two time-scale variance-reduced updates that are correlated with each other. Such an extension of the SVRG scheme to the two time-scale updates is novel and requires new technical developments. Specifically, we need to develop tight variance bounds for the two time-scale updates under Markovian sampling. Second, unlike the convex objective functions of the conventional GTD type of algorithms, the objective function of VR-Greedy-GQ is generally non-convex due to the non-stationary target policy. Hence, we need to develop new techniques to characterize the per-iteration optimization progress towards a stationary point under nonconvexity. In particular, to analyze the two time-scale variance reduction updates of the algorithm, we introduce a 'fine-tuned' Lyapunov function of the form $R_t^m = J(\theta_t^{(m)}) + c_t\|\theta_t^{(m)} - \widetilde{\theta}^{(m)}\|^2$, where the parameter $c_t$ is fine-tuned to cancel other additional quadratic terms $\|\theta_t^{(m)} - \widetilde{\theta}^{(m)}\|^2$ that are implicitly involved in the tracking error terms. The design of this special Lyapunov function is critical to establish the formal convergence of the algorithm. With these technical developments, we are able to establish an improved finite-time convergence rate and sample complexity for VR-Greedy-GQ.

## 1.2 RELATED WORK

**Q-learning and SARSA with function approximation.** The asymptotic convergence of Q-learning and SARSA under linear function approximation were established in Melo et al. (2008); Perkins & Precup (2003), and their finite-time analysis were developed in Zou et al. (2019); Chen et al. (2019). However, these algorithms may diverge in off-policy training Baird (1995). Also, recent works focused on the Markovian setting. Various analysis techniques have been developed to analyze

the finite-time convergence of TD/Q-learning under Markovian samples. Specifically, Wang et al. (2020) developed a multi-step Lyapunov analysis for addressing the biasedness of the stochastic approximation in Q-learning. Srikant & Ying (2019) developed a drift analysis to the linear stochastic approximation problem. Besides the linear function approximation, the finite-time analysis of Q-learning under neural network function approximation is developed in Xu & Gu (2019).

**GTD algorithms.** The GTD2 and TDC algorithms were developed for off-policy TD learning. Their asymptotic convergence was proved in Sutton et al. (2009a;b); Yu (2017), and their finite-time analysis were developed recently in Dalal et al. (2018); Wang et al. (2017); Liu et al. (2015); Gupta et al. (2019); Xu et al. (2019). The Greedy-GQ algorithm is an extension of these algorithms to optimal control and involves nonlinear updates.

**RL with variance reduction:**   Variance reduction techniques have been applied to various RL algorithms. In TD learning, Du et al. (2017) reformulate the MSPBE problem as a convex-concave saddle-point optimization problem and applied SVRG Johnson & Zhang (2013) and SAGA Defazio et al. (2014) to primal-dual batch gradient algorithm. In Korda & La (2015), the variance-reduced TD algorithm was introduced for solving the MSPBE problem, and later Xu et al. (2020) provided a correct non-asymptotic analysis for this algorithm over Markovian samples. Recently, some other works applied the SVRG , SARAH Nguyen et al. (2017) and SPIDER Fang et al. (2018) variance reduction techniques to develop variance-reduced Q-learning algorithms, e.g., Wainwright (2019); Jia et al. (2020). In these works, TD or TDC algorithms are in the form of linear stochastic approximation, and Q-learning has only a single time-scale update. As a comparison, our VR-Greedy-GQ takes nonlinear two time-scale updates to optimization a nonconvex MSPBE.

## 2   PRELIMINARIES: POLICY OPTIMIZATION AND GREEDY-GQ

In this section, we review some preliminaries of reinforcement learning and recap the Greedy-GQ algorithm under linear function approximation.

### 2.1   POLICY OPTIMIZATION IN REINFORCEMENT LEARNING

In reinforcement learning, an agent takes actions to interact with the environment via a Markov Decision Process (MDP). Specifically, an MDP is specified by the tuple $(\mathcal{S}, \mathcal{A}, \mathbf{P}, r, \gamma)$, where $\mathcal{S}$ and $\mathcal{A}$ respectively correspond to the state and action spaces that include finite elements, $r : \mathcal{S} \times \mathcal{A} \times \mathcal{S} \to [0, +\infty)$ denotes a reward function and $\gamma \in (0, 1)$ is the associated reward discount factor.

At any time $t$, assume that the agent is in the state $s_t \in \mathcal{S}$ and takes a certain action $a_t \in \mathcal{A}$ following a stationary policy $\pi$, i.e., $a_t \sim \pi(\cdot|s_t)$. Then, at the subsequent time $t + 1$, the current state of the agent transfers to a new state $s_{t+1}$ according to the transition kernel $\mathbf{P}(\cdot|s_t, a_t)$. At the same time, the agent receives a reward $r_t = r(s_t, a_t, s_{t+1})$ from the environment for this action-state transition. To evaluate the quality of a given policy $\pi$, we often use the action-state value function $Q^\pi : \mathcal{S} \times \mathcal{A} \to \mathbb{R}$ that accumulates the discounted rewards as follows:

$$Q^\pi(s, a) = \mathbb{E}_{s' \sim \mathbf{P}(\cdot|s,a)} \left[ r(s, a, s') + \gamma V^\pi(s') \right],$$

where $V^\pi(s)$ is the state value function defined as $V^\pi(s) = \mathbb{E}\left[ \sum_{t=0}^\infty \gamma^t r_t | s_0 = s \right]$. In particular, define the Bellman operator $T^\pi$ such that $T^\pi Q(s, a) = \mathbb{E}_{s',a'}[r(s, a, s') + \gamma Q(s', a')]$ for any $Q(s, a)$, where $a' \sim \pi(\cdot|s')$. Then, $Q^\pi(s, a)$ is a fixed point of $T^\pi$, i.e.,

$$T^\pi Q^\pi(s, a) = Q^\pi(s, a), \quad \forall s, a. \tag{1}$$

The goal of policy optimization is to learn the optimal policy $\pi^*$ that maximizes the expected total reward $\mathbb{E}[\sum_{t=0}^\infty \gamma^t r_t | s_0 = s]$ for any initial state $s \in \mathcal{S}$, and this is equivalent to learn the optimal value function $Q^*(s, a) = \sup_\pi Q^\pi(s, a), \forall s, a$. In particular, $Q^*$ is a fixed point of the Bellman operator $T$ that is defined as $TQ(s, a) = \mathbb{E}_{s' \sim \mathbf{P}(\cdot|s,a)}[r(s, a, s') + \gamma \max_{b \in \mathcal{A}} Q(s', b)]$.

### 2.2   GREEDY-GQ WITH LINEAR FUNCTION APPROXIMATION

The Greedy-GQ algorithm is inspired by the fixed point characterization in eq. (1), and in the tabular setting it aims to minimize the Bellman error $\|T^\pi Q^\pi - Q^\pi\|_{\mu_{s,a}}^2$. Here, $\| \cdot \|_{\mu_{s,a}}^2$ is induced by

the state-action stationary distribution $\mu_{s,a}$ (induced by the behavior policy $\pi_b$), and is defined as $\|Q\|_{\mu_{s,a}}^2 = \mathbb{E}_{(s,a) \sim \mu_{s,a}}[Q(s,a)^2]$.

In practice, the state and action spaces may include a large number of elements that makes tabular approach infeasible. To address this issue, function approximation technique is widely applied. In this paper, we consider approximating the state-action value function $Q(s,a)$ by a linear function. Specifically, consider a set of basis functions $\{\phi^{(i)} : \mathcal{S} \times \mathcal{A} \to \mathbb{R}, \; i = 1, 2, \ldots, d\}$, each of which maps a given state-action pair to a certain value. Define $\phi_{s,a} = [\phi^{(1)}(s,a); \ldots; \phi^{(d)}(s,a)]$ as the feature vector for $(s,a)$. Then, under linear function approximation, the value function $Q(s,a)$ is approximated by $Q_\theta(s,a) = \phi_{s,a}^\top \theta$, where $\theta \in \mathbb{R}^d$ denotes the parameter of the linear approximation. Consequently, Greedy-GQ aims to find the optimal $\theta^*$ that minimizes the following mean squared projected Bellman error (MSPBE).

$$\text{(MSPBE):} \quad J(\theta) := \frac{1}{2}\|\Pi T^{\pi_\theta} Q_\theta - Q_\theta\|_{\mu_{s,a}}^2, \tag{2}$$

where $\mu_{s,a}$ is the stationary distribution induced by the behavior policy $\pi_b$, $\Pi$ is a projection operator that maps an action-value function $Q$ to the space $\mathcal{Q}$ spanned by the feature vectors, i.e., $\Pi Q = \arg\min_{U \in \mathcal{Q}} \|U - Q\|_{\mu_{s,a}}$. Moreover, the policy $\pi_\theta$ is parameterized by $\theta$. In this paper, we consider the class of Lipschitz and smooth policies (see Assumption 4.2).

Next, we introduce the Greedy-GQ algorithm. Define $\overline{V}_{s'}(\theta) = \sum_{a' \in \mathcal{A}} \pi_\theta(a'|s')\phi_{s',a'}^\top \theta$, $\delta_{s,a,s'}(\theta) = r(s,a,s') + \gamma \overline{V}_{s'}(\theta) - \phi_{s,a}^\top \theta$ and denote $\widehat{\phi}_s(\theta) = \nabla \overline{V}_s(\theta)$. Then, the gradient of the objective function $J(\theta)$ in eq. (2) is expressed as

$$\nabla J(\theta) = -\mathbb{E}[\delta_{s,a,s'}(\theta)\phi_{s,a}] + \gamma \mathbb{E}[\widehat{\phi}_{s'}(\theta)\phi_{s,a}^\top]\omega^*(\theta),$$

where $\omega^*(\theta) = \mathbb{E}[\phi_{s,a}\phi_{s,a}^\top]^{-1}\mathbb{E}[\delta_{s,a,s'}(\theta)\phi_{s,a}]$. To address the double-sampling issue when estimating the product of expectations involved in $\mathbb{E}[\widehat{\phi}_{s'}(\theta)\phi_{s,a}^\top]\omega^*(\theta)$, Sutton et al. (2009a) applies a weight doubling trick and constructs the following two time-scale update rule for the Greedy-GQ algorithm: for every $t = 0, 1, 2, \ldots$, sample $(s_t, a_t, r_t, s_{t+1})$ using the behavior policy $\pi_b$ and do

$$\text{(Greedy-GQ):} \quad \begin{cases} \theta_{t+1} = \theta_t - \eta_\theta\big(-\delta_{t+1}(\theta_t)\phi_t + \gamma(\omega_t^\top \phi_t)\widehat{\phi}_{t+1}(\theta_t)\big), \\ \omega_{t+1} = \omega_t - \eta_\omega\big(\phi_t^\top \omega_t - \delta_{t+1}(\theta_t)\big)\phi_t, \\ \pi_{\theta_{t+1}} = \mathcal{P}(\phi^\top \theta_{t+1}). \end{cases} \tag{3}$$

where $\eta_\theta, \eta_\omega > 0$ are the learning rates and we denote $\delta_{t+1}(\theta) := \delta_{s_t,a_t,s_{t+1}}(\theta)$, $\phi_t := \phi_{s_t,a_t}$, $\widehat{\phi}_{t+1}(\theta_t) := \widehat{\phi}_{s_{t+1}}(\theta_t)$ for simplicity. To elaborate, the first two steps correspond to the two time-scale updates for updating the value function $Q_\theta$, whereas the last step is a policy improvement operation that exploits the updated value function to improve the target policy, e.g., greedy, $\epsilon$-greedy, softmax and mellowmax Asadi & Littman (2017).

The above Greedy-GQ algorithm uses a single Markovian sample to perform the two time-scale updates in each iteration. Such a stochastic Markovian sampling often induces a large variance that significantly slows down the overall convergence. This motivates us to develop variance reduction schemes for the two time-scale Greedy-GQ in the next section.

## 3 GREEDY-GQ WITH VARIANCE REDUCTION

In this section, we propose a variance-reduced Greedy-GQ (VR-Greedy-GQ) algorithm under Markovian sampling by leveraging the SVRG variance reduction scheme Johnson & Zhang (2013). To simplify notations, we define the stochastic updates regarding a sample $x_t = (s_t, a_t, r_t, s_{t+1})$ used in the Greedy-GQ as follows:

$$G_{x_t}(\theta, \omega) := -\delta_{t+1}(\theta)\phi_t + \gamma(\omega^\top \phi_t)\widehat{\phi}_{t+1}(\theta),$$
$$H_{x_t}(\theta, \omega) := \big(\phi_t^\top \omega - \delta_{t+1}(\theta)\big)\phi_t.$$

Next, consider a single MDP trajectory $\{x_t\}_{t \geq 0}$ obtained by the behavior policy $\pi_b$. In particular, we divide the entire trajectory into multiple batches of samples $\{\mathcal{B}_m\}_{m \geq 1}$ so that $\mathcal{B}_m =$

$\{x_{(m-1)M}, ..., x_{mM-1}\}$, and our proposed VR-Greedy-GQ uses one batch of samples in every epoch. To elaborate, in the $m$-th epoch, we first initialize this epoch with a pair of reference points $\theta_0^{(m)} = \widetilde{\theta}^{(m)}, \omega_0^{(m)} = \widetilde{\omega}^{(m)}$, where $\widetilde{\theta}^{(m)}, \widetilde{\omega}^{(m)}$ are set to be the output points $\theta_M^{(m-1)}, \omega_M^{(m-1)}$ of the previous epoch, respectively. Then, we compute a pair of reference batch updates using the reference points and the batch of samples as follows

$$\widetilde{G}^{(m)} = \frac{1}{M} \sum_{k=(m-1)M}^{mM-1} G_{x_k}(\widetilde{\theta}^{(m)}, \widetilde{\omega}^{(m)}), \quad \widetilde{H}^{(m)} = \frac{1}{M} \sum_{k=(m-1)M}^{mM-1} H_{x_k}(\widetilde{\theta}^{(m)}, \widetilde{\omega}^{(m)}). \quad (4)$$

In the $t$-th iteration of the $m$-th epoch, we first query a random sample $x_{\xi_t^m}$ from the batch $\mathcal{B}_m$ uniformly with replacement (i.e., sample $\xi_t^m$ from $\{(m-1)M, ..., mM-1\}$ uniformly). Then, we use this sample to compute the stochastic updates $G_{x_{\xi_t^m}}, H_{x_{\xi_t^m}}$ at both of the points $(\theta_t^{(m)}, \omega_t^{(m)}), (\widetilde{\theta}^{(m)}, \widetilde{\omega}^{(m)})$. After that, we use these stochastic updates and the reference batch updates to construct the variance-reduced updates in Algorithm 1 via the SVRG scheme, where for simplicity we denote the stochastic updates $G_{x_{\xi_t^m}}, H_{x_{\xi_t^m}}$ respectively as $G_t^{(m)}, H_t^{(m)}$. In particular, we project the two time-scale updates onto the Euclidean ball with radius $R$ to stabilize the algorithm updates, and we assume that $R$ is large enough to include at least one stationary point of $J$. Lastly, we further update the policy via the policy improvement operation $\mathcal{P}$.

---
**Algorithm 1:** Variance-Reduced Greedy-GQ

**Input:** learning rates $\eta_\theta, \eta_\omega$, batch size $M$.
**Initialize:** $\widetilde{\theta}^{(1)} = \theta_0, \widetilde{\omega}^{(1)} = \omega_0, \pi_{\widetilde{\theta}^{(1)}} \leftarrow \mathcal{P}(\phi^\top \widetilde{\theta}^{(1)})$.
**for** $m = 1, 2, \ldots$ **do**
   $\theta_0^{(m)} = \widetilde{\theta}^{(m)}, \omega_0^{(m)} = \widetilde{\omega}^{(m)}$. Compute $\widetilde{G}^{(m)}, \widetilde{H}^{(m)}$ according to eq. (4).
   **for** $t = 0, 1, \ldots, M - 1$ **do**
      Query a sample from $\mathcal{B}_m$ with replacement.
      $\theta_{t+1}^{(m)} = \Pi_R \left[ \theta_t^{(m)} - \eta_\theta \left( G_t^{(m)}(\theta_t^{(m)}, \omega_t^{(m)}) - G_t^{(m)}(\widetilde{\theta}^{(m)}, \widetilde{\omega}^{(m)}) + \widetilde{G}^{(m)} \right) \right]$.
      $\omega_{t+1}^{(m)} = \Pi_R \left[ \omega_t^{(m)} - \eta_\omega \left( H_t^{(m)}(\theta_t^{(m)}, \omega_t^{(m)}) - H_t^{(m)}(\tilde{\theta}^{(m)}, \tilde{\omega}^{(m)}) + \widetilde{H}^{(m)} \right) \right]$.
      **Policy improvement**: $\pi_{\theta_{t+1}^{(m)}} \leftarrow \mathcal{P}(\phi^\top \theta_{t+1}^{(m)})$.
   **end**
   Set $\widetilde{\theta}^{(m+1)} = \theta_M^{(m)}, \widetilde{\omega}^{(m+1)} = \omega_M^{(m)}$.
**end**
**Output:** parameter $\theta$ chosen among $\{\theta_t^{(m)}\}_{t,m}$ uniformly at random.

---

The above VR-Greedy-GQ algorithm has several advantages and uniqueness. First, it takes incremental updates that use a single Markovian sample per-iteration. This makes the algorithm sample efficient. Second, VR-Greedy-GQ applies variance reduction to both of the two time-scale updates. As we show later in the analysis, such a two time-scale variance reduction scheme significantly reduces the variance error of both of the stochastic updates.

We want to further clarify the incrementalism and online property of VR-Greedy-GQ. Our VR-Greedy-GQ is based on the online-SVRG and can be viewed as an incremental algorithm with regard to the batches of samples used in the outer-loops, i.e., in every outer-loop the algorithm samples a new batch of samples and use them to perform variance reduction in the corresponding inner-loops. Therefore, VR-Greedy-GQ can be viewed as an online batch-incremental algorithm. In general, there is a trade-off between incrementalism and variance reduction for SVRG-type algorithms: a larger batch size in the outer-loops enhances the effect of variance reduction, while a smaller batch size makes the algorithm more incremental.

## 4 FINITE-TIME ANALYSIS OF VR-GREEDY-GQ

In this section, we analyze the finite-time convergence rate of VR-Greedy-GQ. We adopt the following standard technical assumptions from Wang & Zou (2020); Xu et al. (2020).

**Assumption 4.1** (Feature boundedness). *The feature vectors are uniformly bounded, i.e., $\|\phi_{s,a}\| \le 1$ for all $(s,a) \in \mathcal{S} \times \mathcal{A}$.*

**Assumption 4.2** (Policy smoothness). *The mapping $\theta \mapsto \pi_\theta$ is $k_1$-Lipschitz and $k_2$-smooth.*

We note that the above class of smooth policies covers a variety of practical policies, including softmax and mellowmax policies Asadi & Littman (2017); Wang & Zou (2020).

**Assumption 4.3** (Problem solvability). *The matrix $C := \mathbb{E}[\phi_{s,a}\phi_{s,a}^\top]$ is non-singular.*

**Assumption 4.4** (Geometric uniform ergodicity). *There exists $\Lambda > 0$ and $\rho \in (0,1)$ such that*

$$\sup_{s \in \mathcal{S}} d_{TV}\big(\mathbb{P}(s_t|s_0 = s), \mu\big) \le \Lambda \rho^t,$$

*for any $t > 0$, where $d_{TV}$ is the total-variation distance.*

Based on the above assumptions, we obtain the following finite-time convergence rate result.

**Theorem 4.5** (Finite-time convergence). *Let Assumptions 4.1– 4.4 hold and consider the VR-Greedy-GQ algorithm. Choose learning rates $\eta_\theta$, $\eta_\omega$ and the batch size $M$ that satisfy the conditions specified in eqs. (15) to (19). Then, after $T$ epochs, the output of the algorithm satisfies*

$$\mathbb{E}\|\nabla J(\theta_\xi^{(\zeta)})\|^2 \le \mathcal{O}\Big(\frac{1}{\eta_\theta TM} + \frac{1}{T}\big(\eta_\omega + \frac{\eta_\theta^2}{\eta_\omega^2}\big) + \big(\eta_\omega + \frac{\eta_\theta^2}{\eta_\omega^2}\big)^2 + \frac{1}{M}\Big),$$

*where $\xi, \zeta$ are random indexes that are sampled from $\{0, ..., M-1\}$ and $\{1, ..., T\}$ uniformly at random, respectively.*

Theorem 4.5 shows that VR-Greedy-GQ asymptotically converges to a neighborhood of a stationary point at a sublinear rate. In particular, the size of the neighborhood is in the order of $\mathcal{O}(M^{-1} + \eta_\theta^4 \eta_\omega^{-4} + \eta_\omega^2)$, which can be driven arbitrarily close to zero by choosing a large batch size and sufficiently small learning rates that satisfy the two time-scale condition $\eta_\theta/\eta_\omega \to 0$. Moreover, the convergence error terms implicitly include a bias error $\mathcal{O}(\frac{1}{M})$ caused by the Markovian sampling and a variance error $\mathcal{O}(\frac{\eta_\theta}{M})$ caused by the stochastic updates, both of which are substantially reduced by the large batch size $M$. This shows that the SVRG scheme can effectively reduce the bias and variance error of the two time-scale stochastic updates.

By further optimizing the choice of hyper-parameters, we obtain the following characterization of sample complexity of VR-Greedy-GQ.

**Corollary 4.6** (Sample complexity). *Under the same conditions as those of Theorem 4.5, choose learning rates so that $\eta_\theta = \mathcal{O}(\frac{1}{M})$, $\eta_\omega = \mathcal{O}(\eta_\theta^{2/3})$, and set $T,M = \mathcal{O}(\epsilon^{-1})$). Then, the required sample complexity for achieving $\mathbb{E}\|\nabla J(\theta_\xi^{(\zeta)})\|^2 \le \epsilon$ is in the order of $TM = \mathcal{O}(\epsilon^{-2})$.*

Such a complexity result is orderwise lower than the complexity $\mathcal{O}(\epsilon^{-3})$ of the original Greedy-GQ Wang & Zou (2020). Therefore, this demonstrates the advantage of applying variance reduction to the two time-scale updates of VR-Greedy-GQ. We also note that for online stochastic non-convex optimization, the sample complexity of the SVRG algorithm is in the order of $\mathcal{O}(\epsilon^{-5/3})$ Li & Li (2018), which is slightly better than our result. This is reasonable as the SVRG in stochastic optimization is unbiased due to the i.i.d. sampling. In comparison, VR-Greedy-GQ works on a single MDP trajectory that induces Markovian noise, and the two-timescale updates of the algorithm also introduces additional tracking error.

## 5 SKETCH OF THE TECHNICAL PROOF

In this section, we provide an outline of the technical proof of the main Theorem 4.5 and highlight the main technical contributions. The details of the proof can be found in the appendix.

We note that our proof logic partly follows the that of the conventional SVRG, i.e., exploiting the objective function smoothness and introducing a Lyapunov function. However, our analysis requires substantial new developments to address the challenges of off-policy control, two time-scale updates of VR-Greedy-GQ and correlation of Markovian samples.

The key step of the proof is to develop a proper Lyapunov function that drives the parameter to a stationary point along the iterations. In addition, we also need to develop tight bounds for the bias error, variance error and tracking error. We elaborate the key steps of the proof below.

**Step 1:** We first define the following Lyapunov function with certain $c_t > 0$ to be determined later.

$$R_t^m := J(\theta_t^{(m)}) + c_t \|\theta_t^{(m)} - \widetilde{\theta}^{(m)}\|^2. \tag{5}$$

To explain the motivation, note that unlike the analysis of variance-reduced TD learning Xu et al. (2020) where the term $\|\theta_t^{(m)} - \widetilde{\theta}^{(m)}\|^2$ can be decomposed into $\|\theta_t^{(m)} - \theta^*\|^2 + \|\widetilde{\theta}^{(m)} - \theta^*\|^2$ to get the desired upper bound, here we do not have $\theta^*$ due to the non-convexity of $J(\theta)$. Hence, we need to properly merge this term into the Lyapunov function $R_t^m$. By leveraging the smoothness of $J(\theta)$ and the algorithm update rule, we obtain the following bound for the Lyapunov function $R_t^m$ (see eq. (10) in the appendix for the details).

$$\mathbb{E}[R_{t+1}^m] \leq \mathbb{E}[J(\theta_t^{(m)})] + \mathcal{O}\big(\mathbb{E}[\|\theta_t^{(m)} - \widetilde{\theta}^{(m)}\|^2] + \mathbb{E}[\|\nabla J(\theta_t^{(m)})\|]\|^2 + M^{-1}\big)$$
$$+ \mathcal{O}\big(\mathbb{E}[\|\widetilde{\omega}^{(m)} - \omega^*(\widetilde{\theta}^{(m)})\|^2] + \mathbb{E}[\|\omega_t^{(m)} - \omega^*(\theta_t^{(m)})\|^2]\big). \tag{6}$$

In particular, the error term $\frac{1}{M}$ is due to the noise of Markovian sampling and the variance of the stochastic updates, and the last two terms correspond to tracking errors.

**Step 2:** To telescope the Lyapunov function over $t$ based on eq. (6), one may want to define $J(\theta_t^{(m)})] + \mathcal{O}\big(\mathbb{E}[\|\theta_t^{(m)} - \widetilde{\theta}^{(m)}\|^2]\big) = R_t^m$ by choosing a proper $c_t$ of $R_t^m$. However, note that eq. (6) involves the last two tracking error terms, which also implicitly depend on $\mathbb{E}\|\theta_t^{(m)} - \widetilde{\theta}^{(m)}\|^2$ as we show later in the Step 3. Therefore, we need to carefully define the $c_t$ of $R_t^m$ so that after applying the tracking error bounds developed in the Step 3, the right hand side of eq. (6) can yield an $R_t^m$ without involving the term $\mathbb{E}\|\theta_t^{(m)} - \widetilde{\theta}^{(m)}\|^2$. It turns out that we need to define $c_t$ via the recursion specified in eq. (11) in the appendix. We rigorously show that the sequence $\{c_t\}_t$ is uniformly bounded by a small constant $\widehat{c} = \frac{1}{8}$. Then, plugging these bounds into eq. (6) and summing over one epoch, we obtain the following bound (see eq. (13) in the appendix for the details).

$$\eta_\theta \sum_{t=0}^{M-1} \mathbb{E}[\|\nabla J(\theta_t^{(m)})\|^2] \leq \mathbb{E}[R_0^m] - \mathbb{E}[R_M^m] + \mathcal{O}\big(\eta_\theta + \eta_\theta^2 M \mathbb{E}[\|\widetilde{\omega}^{(m)} - \omega^*(\widetilde{\theta}^{(m)})\|^2]\big)$$
$$+ \mathcal{O}\Big(\eta_\theta \sum_{t=0}^{M-1} \mathbb{E}\|\omega_t^{(m)} - \omega^*(\theta_t^{(m)})\|^2 - \eta_\theta \widehat{c}\Big(\eta_\omega + \frac{\eta_\theta^2}{\eta_\omega^2}\Big) \sum_{t=0}^{M-1} \mathbb{E}[\|\theta_t^{(m)} - \widetilde{\theta}^{(m)}\|^2]\Big).$$

**Step 3:** We derive bounds for the tracking error terms $\sum_{t=0}^{M-1} \mathbb{E}\|\omega_t^{(m)} - \omega^*(\theta_t^{(m)})\|^2$ and $\mathbb{E}\|\widetilde{\omega}^{(m)} - \omega^*(\widetilde{\theta}^{(m)})\|^2$ in the above bound in Lemma D.7 and Lemma D.8.

**Step 4:** Lastly, by substituting the tracking error bounds obtained in Step 3 into the bound obtained in Step 2, the resulting bound does not involve the term $\sum_{t=0}^{M-1} \mathbb{E}\|\theta_t^{(m)} - \widetilde{\theta}^{(m)}\|^2$. Then, summing this bound over the epochs $m = 1, ..., T$, we obtain the desired finite-time convergence rate result.

## 6  EXPERIMENTS

In this section, we conduct two reinforcement learning experiments, namely, Garnet problem Archibald et al. (1995) and Frozen Lake game Brockman et al. (2016), to test the performance of VR-Greedy-GQ in the off-policy setting, and compare it with Greedy-GQ in the Markovian setting.

### 6.1  GARNET PROBLEM

For the Garnet problem, we refer to Appendix F for the details of the problem setup. In Figure 1 (left), we plot the minimum gradient norm *v.s.* the number of pseudo stochastic gradient computations for both algorithms using 40 Garnet MDP trajectories, and each trajectory contains 10k samples. The upper and lower envelopes of the curves correspond to the $95\%$ and $5\%$ percentiles of the 40 curves, respectively. It can be seen that VR-Greedy-GQ outperforms Greedy-GQ and achieves a significantly smaller asymptotic gradient norm.

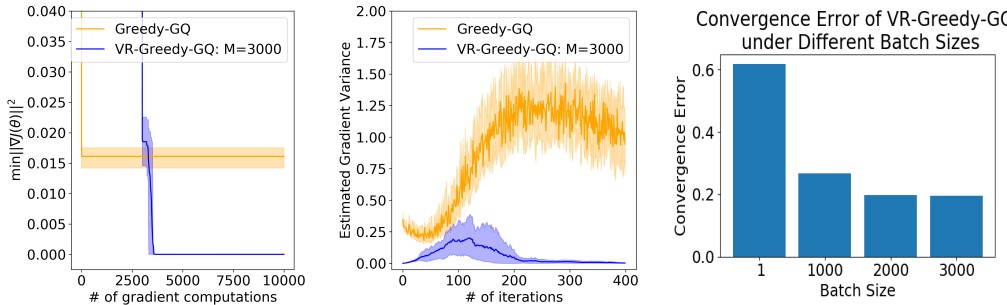

Figure 1: Comparison of Greedy-GQ and VR-Greedy-GQ in solving the Garnet problem.

In Figure 1 (middle), we track the estimated variance of the stochastic update for both algorithms along the iterations. Specifically, we query 500 Monte Carlo samples per iteration to estimate the pseudo gradient variance $\mathbb{E}\|G_t^{(m)}(\theta_t^{(m)}, \omega_t^{(m)}) - \nabla J(\theta_t^{(m)})\|^2$. It can be seen from the figure that the stochastic updates of VR-Greedy-GQ induce a much smaller variance than Greedy-GQ. This demonstrates the effectiveness of the two time-scale variance reduction scheme of VR-Greedy-GQ.

We further study the asymptotic convergence error of VR-Greedy-GQ under different batch sizes $M$. We use the default learning rate setting that is mentioned previously and run 100k iterations for one Garnet trajectories. We use the mean of the convergence error of the last 10k iterations as an estimate of the asymptotic convergence error (the training curves are already saturated and flattened). Figure 1 (right) shows the asymptotic convergence error of VR-Greedy-GQ under different batch sizes $M$. It can be seen that VR-Greedy-GQ achieves a smaller asymptotic convergence error with a larger batch size, which matches our theoretical result.

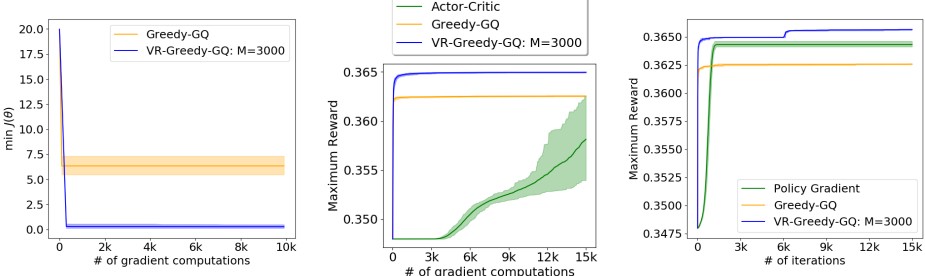

Figure 2: Comparison of MSPBE and reward obtained by Greedy-GQ, VR-Greedy-GQ and PG.

In Figure 2 (Left), we plot the MSPBE $J(\theta)$ *v.s.* number of gradient computations for both Greedy-GQ and VR-Greedy-GQ, where one can see that VR-Greedy-GQ achieves a much smaller MSPBE than Greedy-GQ. In Figure 2 (Middle), we plot the estimated expected maximum reward (see Appendix F for details) *v.s.* number of gradient computations for Greedy-GQ, VR-Greedy-GQ and actor-critic, where for actor-critic we set learning rate $\eta_\theta = 0.02$ for the actor update and $\eta_\omega = 0.01$ for the critic update. One can see that VR-Greedy-GQ achieves a higher reward than the other two algorithms, demonstrating the high quality of its learned policy. In addition, we also plot the estimated expected maximum reward *v.s.* number of iterations for Greedy-GQ, VR-Greedy-GQ and policy gradient in Figure 2 (Right). For the policy gradient, we apply the standard off-policy policy gradient algorithm. For each update, we sample 30 independent trajectories with a fixed length 60 to estimate the expected discounted return. The learning rate of policy gradient is set as $\eta_\theta$. We note that each iteration of policy gradient consumes 1800 samples and hence it is very sample inefficient. Hence we set the $x$-axis to be number of iterations for a clear presentation (otherwise it becomes a flat curve). One can see that VR-Greedy-GQ achieves a much higher expected reward than both Greedy-GQ and policy gradient.

## 6.2 FROZEN LAKE GAME

We further test these algorithms in solving the more complex frozen lake game. we refer to Appendix F for the details of the problem setup. Figure 3 shows the comparison between VR-Greedy-

GQ and Greedy-GQ, and one can make consistent observations with those made in the Garnet experiment. Specifically, Figure 3 (left) shows that VR-Greedy-GQ achieves a much more stationary policy than Greedy-GQ. Figure 3 (middle) shows that the stochastic updates of VR-Greedy-GQ induce a much smaller variance than those of Greedy-GQ. Moreover, Figure 3 (right) verifies our theoretical result that VR-Greedy achieves a smaller asymptotic convergence error with a larger batch size.

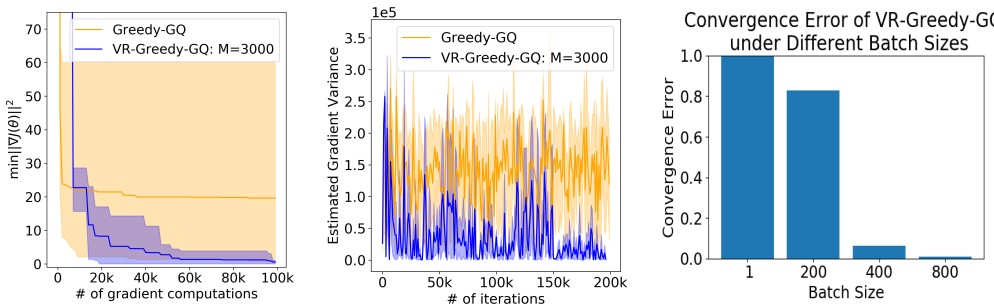

Figure 3: Comparison of Greedy-GQ and VR-Greedy-GQ in solving the Frozen Lake problem.

We further plot the MSPBE *v.s.* number of gradient computations for both Greedy-GQ and VR-Greedy-GQ in Figure 4 (Left), where one can see that VR-Greedy-GQ outperforms Greedy-GQ. In Figure 2 (Middle), we plot the estimated expected maximum reward *v.s.* number of gradient computations for Greedy-GQ, VR-Greedy-GQ and actor-critic, where for actor-critic we set learning rate $\eta_\theta = 0.2$ for the actor update and $\eta_\omega = 0.1$ for the critic update. It can be seen that VR-Greedy-GQ achieves a higher reward than the other two algorithms. In Figure 2 (Right), we plot the estimated expected maximum reward *v.s.* number of iterations for Greedy-GQ, VR-Greedy-GQ and policy gradient. For policy gradient, we use the same parameter settings as before. One can see that VR-Greedy-GQ achieves a much higher expected reward than both Greedy-GQ and policy gradient.

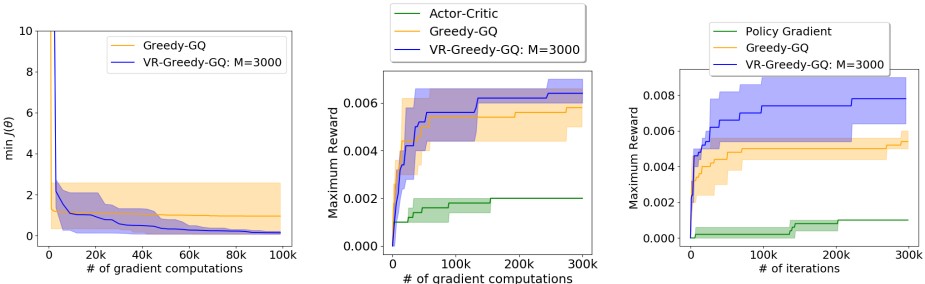

Figure 4: Comparison of MSPBE and reward obtained by Greedy-GQ, VR-Greedy-GQ and PG.

# 7 CONCLUSION

In this paper, we develop a variance-reduced two time-scale Greedy-GQ algorithm for optimal control by leveraging the SVRG variance reduction scheme. Under linear function approximation and Markovian sampling, we establish the sublinear finite-time convergence rate of the algorithm to a stationary point and prove an improved sample complexity bound over that of the original Greedy-GQ. The RL experiments well demonstrated the effectiveness of the proposed two time-scale variance reduction scheme. Our algorithm design may inspire new developments of variance reduction for two time-scale RL algorithms. In the future, we will explore Greedy-GQ with other nonconvex variance reduction schemes to possibly further improve the sample complexity.

## ACKNOWLEDGEMENT

The work of S. Zou was supported by the National Science Foundation under Grant CCF-2007783.

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

# Appendix

## Table of Contents

## A   FILTRATION AND LIST OF CONSTANTS

**Filtration**   We follow the definition of filtration in VRTD (Appendix D, Xu et al. (2020)). Recall that $B_m$ denotes the set of Markovian samples used in the $m$-th epoch, and we also abuse the notation here by letting $x_t^{(m)}$ be the sample picked in the $t$-th iteration of the $m$-th epoch. Then, we define the filtration for Markovian samples as follows

$$F_{1,0} = \sigma(B_0 \cup \sigma(\tilde{\theta}^{(0)}, \tilde{w}^{(0)})), F_{1,1} = \sigma(F_{1,0} \cup \sigma(x_0^{(1)})), \ldots, F_{1,M} = \sigma(F_{1,M-1} \cup \sigma(x_{M-1}^{(1)}))$$

$$F_{2,0} = \sigma(B_1 \cup F_{1,M} \cup \sigma(\tilde{\theta}^{(1)}, \tilde{w}^{(1)})), F_{2,1} = \sigma(F_{2,0} \cup \sigma(x_0^{(2)})), \ldots, F_{2,M} = \sigma(F_{2,M-1} \cup \sigma(x_{M-1}^{(2)}))$$

$$\vdots$$

$$F_{m,0} = \sigma(B_{m-1} \cup F_{m-1,M} \cup \sigma(\tilde{\theta}^{(m-1)}, \tilde{w}^{(m-1)})), F_{m,1} = \sigma(F_{m,0} \cup \sigma(x_0^{(m)})), \ldots,$$

$$F_{m,M} = \sigma(F_{m,M-1} \cup \sigma(x_{M-1}^{(m)})).$$

Moreover, we define $\mathbb{E}_{t,m}$ as the conditional expectation with respect to the $\sigma$-field $F_{t,m}$.

**List of Constants**   We summarize all the constants that are used in the proof as follows.

- $G = r_{\max} + (1+\gamma)R + \gamma(|\mathcal{A}|Rk_1 + 1)R$.
- $H = (2+\gamma)R + r_{\max}$.
- $C_1 = (1 + 2\Lambda\frac{\rho}{1-\rho})(G + C_{\nabla J})^2$.
- $C_2 = H^2(1 + \Lambda\frac{\rho}{1-\rho})$.
- $C_3 = \frac{8R^2}{\lambda_C}(1 + \frac{\rho\Lambda}{1-\rho})$.
- $C_4 = \frac{2}{\lambda_C}(R(2+\gamma) + r_{\max})^2(1 + \frac{\rho\Lambda}{1-\rho})$.

## B   PROOF OF THEOREM 4.5

We first define the following Lyapunov function

$$R_t^m := J(\theta_t^{(m)}) + c_t\|\theta_t^{(m)} - \widetilde{\theta}^{(m)}\|^2,$$

where $c_t > 0$ is to be determined later. Our strategy is to characterize the per-iteration progress of $R_t^m$. In particular, we use Lemma E.9 to bound the first term of $R_t^m$ and use Lemma E.10 to bound the second term of $R_t^m$. Note that Lemma E.9 implies that

$$J(\theta_{t+1}^{(m)}) \leq J(\theta_t^{(m)}) - \eta_\theta \langle \nabla J(\theta_t^{(m)}), g_t^{(m)} - \nabla J(\theta_t^{(m)}) \rangle - \eta_\theta \|\nabla J(\theta_t^{(m)})\|^2 + \frac{L}{2}\eta_\theta^2 \|g_t^{(m)}\|^2.$$

Let $\xi_t^{(m)} := \langle \nabla J(\theta_t^{(m)}), g_t^{(m)} - \nabla J(\theta_t^{(m)}) \rangle$. Then, we obtain that

$$J(\theta_{t+1}^{(m)}) \leq J(\theta_t^{(m)}) - \eta_\theta \xi_t^{(m)} - \eta_\theta \|\nabla J(\theta_t^{(m)})\|^2 + \frac{L}{2}\eta_\theta^2 \|g_t^{(m)}\|^2. \tag{7}$$

Substituting eq. (7) and eq. (30) into the definition of $R_{t+1}^m$, we obtain that

$$\begin{aligned}
R_{t+1}^m &:= J(\theta_{t+1}^{(m)}) + c_{t+1}\|\theta_{t+1}^{(m)} - \widetilde{\theta}^{(m)}\|^2 \\
&\leq J(\theta_t^{(m)}) - \eta_\theta \xi_t^{(m)} - \eta_\theta \|\nabla J(\theta_t^{(m)})\|^2 + \frac{L}{2}\eta_\theta^2 \|g_t^{(m)}\|^2 \\
&\quad + c_{t+1}\Big[\eta_\theta^2 \|g_t^{(m)}\|^2 + \|\theta_t^{(m)} - \widetilde{\theta}^{(m)}\|^2 - 2\eta_\theta \zeta_t^{(m)} \\
&\quad + \eta_\theta\big[\frac{1}{\beta_t}\|\nabla J(\theta_t^{(m)})\|^2 + \beta_t \|\theta_t^{(m)} - \widetilde{\theta}^{(m)}\|^2\big]\Big] \\
&= J(\theta_t^{(m)}) + c_{t+1}(\eta_\theta \beta_t + 1)\|\theta_t^{(m)} - \widetilde{\theta}^{(m)}\|^2 \\
&\quad + \big(-\eta_\theta + c_{t+1}\frac{\eta_\theta}{\beta_t}\big)\|\nabla J(\theta_t^{(m)})\|^2 + \big(\frac{L}{2}\eta_\theta^2 + c_{t+1}\eta_\theta^2\big)\|g_t^{(m)}\|^2 \\
&\quad - \eta_\theta \xi_t^{(m)} - 2c_{t+1}\eta_\theta \zeta_t^{(m)}.
\end{aligned}$$

Next, we bound the two inner product terms $\xi_t^{(m)}$ and $\zeta_t^{(m)}$.

**Bounding the term $\xi_t^{(m)}$:**

$$\xi_t^{(m)} := \langle \nabla J(\theta_t^{(m)}), g_t^{(m)} - \nabla J(\theta_t^{(m)}) \rangle.$$

Recall the variance-reduced stochastic update

$$g_t^{(m)} = G_t^{(m)}(\theta_t^{(m)}, \omega_t^{(m)}) - G_t^{(m)}(\widetilde{\theta}^{(m)}, \widetilde{\omega}^{(m)}) + \widetilde{G}^{(m)}.$$

Then, the term $\xi_t^{(m)}$ can be decomposed as

$$\begin{aligned}
\xi_t^{(m)} &= \langle \nabla J(\theta_t^{(m)}), g_t^{(m)} - \nabla J(\theta_t^{(m)}) \rangle \\
&= \langle \nabla J(\theta_t^{(m)}), G_t^{(m)}(\theta_t^{(m)}, \omega^*(\theta_t^{(m)})) - \nabla J(\theta_t^{(m)}) \rangle \\
&\quad + \langle \nabla J(\theta_t^{(m)}), G_t^{(m)}(\theta_t^{(m)}, \omega_t^{(m)}) - G_t^{(m)}(\theta_t^{(m)}, \omega^*(\theta_t^{(m)})) \rangle \\
&\quad + \langle \nabla J(\theta_t^{(m)}), -G_t^{(m)}(\widetilde{\theta}^{(m)}, \widetilde{\omega}^{(m)}) + \widetilde{G}^{(m)} \rangle
\end{aligned}$$

In the last equality, the first inner product term is the bias caused by Markovian samples, and by Lemma D.3 we have that

$$\begin{aligned}
&\mathbb{E}\langle \nabla J(\theta_t^{(m)}), G_t^{(m)}(\theta_t^{(m)}, \omega^*(\theta_t^{(m)})) - \nabla J(\theta_t^{(m)}) \rangle \\
=&\mathbb{E}\langle \nabla J(\theta_t^{(m)}), G^{(m)}(\theta_t^{(m)}, \omega^*(\theta_t^{(m)})) - \nabla J(\theta_t^{(m)}) \rangle \\
=&\frac{1}{4}\mathbb{E}\|\nabla J(\theta_t^{(m)})\|^2 + \mathbb{E}\|G^{(m)}(\theta_t^{(m)}, \omega^*(\theta_t^{(m)})) - \nabla J(\theta_t^{(m)})\rangle\|^2 \\
\leq&\frac{1}{4}\mathbb{E}\|\nabla J(\theta_t^{(m)})\|^2 + \frac{C_1}{M}.
\end{aligned}$$

The second inner product term is the bias caused by tracking error, and we further obtain that

$$\begin{aligned}
&\langle \nabla J(\theta_t^{(m)}), G_t^{(m)}(\theta_t^{(m)}, \omega_t^{(m)}) - G_t^{(m)}(\theta_t^{(m)}, \omega^*(\theta_t^{(m)})) \rangle \\
\leq&\frac{1}{4}\|\nabla J(\theta_t^{(m)})\|^2 + L_1\|\omega_t^{(m)} - \omega^*(\theta_t^{(m)})\|^2.
\end{aligned}$$

The third inner product term is unbiased. Combining all of these bounds, we finally obtain that

$$|\mathbb{E}\xi_t^{(m)}| \leq \frac{C_1}{M} + \frac{1}{2}\|\nabla J(\theta_t^{(m)})\|^2 + L_1\|\omega_t^{(m)} - \omega^*(\theta_t^{(m)})\|^2. \tag{8}$$

**Bounding the term $\zeta_t^{(m)}$:**

$$\zeta_t^{(m)} := \langle g_t^{(m)} - \nabla J(\theta_t^{(m)}), \theta_t^{(m)} - \widetilde{\theta}^{(m)} \rangle.$$

Similar to the previous proof for bounding $\xi_t^{(m)}$, we can decompose $\zeta_t^{(m)}$ as

$$
\begin{aligned}
\zeta_t^{(m)} &= \langle \theta_t^{(m)} - \widetilde{\theta}^{(m)}, g_t^{(m)} - \nabla J(\theta_t^{(m)}) \rangle \\
&= \langle \theta_t^{(m)} - \widetilde{\theta}^{(m)}, G_t^{(m)}(\theta_t^{(m)}, \omega^*(\theta_t^{(m)})) - \nabla J(\theta_t^{(m)}) \rangle \\
&\quad + \langle \theta_t^{(m)} - \widetilde{\theta}^{(m)}, G_t^{(m)}(\theta_t^{(m)}, \omega_t^{(m)}) - G_t^{(m)}(\theta_t^{(m)}, \omega^*(\theta_t^{(m)})) \rangle \\
&\quad + \langle \theta_t^{(m)} - \widetilde{\theta}^{(m)}, -G_t^{(m)}(\widetilde{\theta}^{(m)}, \widetilde{\omega}^{(m)}) + \widetilde{G}^{(m)} \rangle
\end{aligned}
$$

In the last equality, the first inner product term is the bias caused by Markovian samples. We obtain that

$$
\begin{aligned}
&\mathbb{E}\langle \theta_t^{(m)} - \widetilde{\theta}^{(m)}, G_t^{(m)}(\theta_t^{(m)}, \omega^*(\theta_t^{(m)})) - \nabla J(\theta_t^{(m)}) \rangle \\
&\leq \frac{1}{2}\mathbb{E}\|\theta_t^{(m)} - \widetilde{\theta}^{(m)}\|^2 + \frac{1}{2}\mathbb{E}\|G_t^{(m)}(\theta_t^{(m)}, \omega^*(\theta_t^{(m)})) - \nabla J(\theta_t^{(m)})\rangle\|^2 \\
&\leq \frac{1}{2}\mathbb{E}\|\theta_t^{(m)} - \widetilde{\theta}^{(m)}\|^2 + \frac{1}{2}\frac{C_1}{M}.
\end{aligned}
$$

The second inner product term is the bias caused by tracking error. We obtain that

$$
\begin{aligned}
&\langle \theta_t^{(m)} - \widetilde{\theta}^{(m)}, G_t^{(m)}(\theta_t^{(m)}, \omega_t^{(m)}) - G_t^{(m)}(\theta_t^{(m)}, \omega^*(\theta_t^{(m)})) \rangle \\
&\leq \frac{1}{2}\|\theta_t^{(m)} - \widetilde{\theta}^{(m)}\|^2 + \frac{L_1}{2}\|\omega_t^{(m)} - \omega^*(\theta_t^{(m)})\|^2.
\end{aligned}
$$

The third inner product term is unbiased. Combining all of these bounds, we finally obtain that

$$|\mathbb{E}\zeta_t^{(m)}| \leq \frac{1}{2}\frac{C_1}{M} + \|\theta_t^{(m)} - \widetilde{\theta}^{(m)}\|^2 + \frac{L_1}{2}\|\omega_t^{(m)} - \omega^*(\theta_t^{(m)})\|^2. \tag{9}$$

Next, we continue to bound the Lyapunov function. Recall we have shown that

$$
\begin{aligned}
R_{t+1}^m &\leq J(\theta_t^{(m)}) + c_{t+1}(\eta_\theta\beta_t + 1)\|\theta_t^{(m)} - \widetilde{\theta}^{(m)}\|^2 \\
&\quad + \big(-\eta_\theta + c_{t+1}\frac{\eta_\theta}{\beta_t}\big)\|\nabla J(\theta_t^{(m)})\|^2 + \big(\frac{L}{2}\eta_\theta^2 + c_{t+1}\eta_\theta^2\big)\|g_t^{(m)}\|^2 \\
&\quad - \eta_\theta\xi_t^{(m)} - 2c_{t+1}\eta_\theta\zeta_t^{(m)}.
\end{aligned}
$$

Taking expectation on both sides of the above inequality and applying eq. (8), eq. (9), and Lemma D.1, we obtain that

$$
\begin{aligned}
\mathbb{E}[R_{t+1}^m] &\leq \mathbb{E}\big[J(\theta_t^{(m)}) + c_{t+1}(\eta_\theta\beta_t + 1)\|\theta_t^{(m)} - \widetilde{\theta}^{(m)}\|^2\big] \\
&\quad + \big(-\eta_\theta + c_{t+1}\frac{\eta_\theta}{\beta_t}\big)\mathbb{E}\|\nabla J(\theta_t^{(m)})\|^2 \\
&\quad + \big(\frac{L}{2}\eta_\theta^2 + c_{t+1}\eta_\theta^2\big)\Big[6L_1\mathbb{E}\|\omega_t^{(m)} - \omega^*(\theta_t^{(m)})\|^2 + 9L_1\mathbb{E}\|\widetilde{\omega}^{(m)} - \omega^*(\widetilde{\theta}^{(m)})\|^2 \\
&\quad + 9L_2\mathbb{E}\|\theta_t^{(m)} - \widetilde{\theta}^{(m)}\|^2 + \frac{1}{M}\cdot 9C_1 + 9\mathbb{E}\|\nabla J(\theta_t^{(m)})\|^2\Big] \\
&\quad + \eta_\theta\big[\frac{C_1}{M} + \frac{1}{2}\|\nabla J(\theta_t^{(m)})\|^2 + L_1\|\omega_t^{(m)} - \omega^*(\theta_t^{(m)})\|^2\big] \\
&\quad + 2c_{t+1}\eta_\theta\big[\frac{1}{2}\frac{C_1}{M} + \|\theta_t^{(m)} - \widetilde{\theta}^{(m)}\|^2 + \frac{L_1}{2}\|\omega_t^{(m)} - \omega^*(\theta_t^{(m)})\|^2\big]. \tag{10}
\end{aligned}
$$

We note that the tracking error term $\|\omega_t^{(m)} - \omega^*(\theta_t^{(m)})\|^2$ has dependence on $\|\theta_t^{(m)} - \widetilde{\theta}^{(m)}\|^2$. Here we use a trick to merge this dependence to the coefficient $c_{t+1}$. Specifically, we add and subtract the same term in the above bound and obtain that

$$
\begin{aligned}
\mathbb{E}[R_{t+1}^m] \leq \mathbb{E}\Big[ & J(\theta_t^{(m)}) + \big[c_{t+1}(\eta_\theta \beta_t + 1 + 2\eta_\theta) + 9L_1\big(\tfrac{L}{2}\eta_\theta^2 + c_{t+1}\eta_\theta^2\big)\big]\|\theta_t^{(m)} - \widetilde{\theta}^{(m)}\|^2\Big] \\
& + \Big[-\tfrac{1}{2}\eta_\theta + c_{t+1}\tfrac{\eta_\theta}{\beta_t} + 9\big(\tfrac{L}{2}\eta_\theta^2 + c_{t+1}\eta_\theta^2\big)\Big]\mathbb{E}\|\nabla J(\theta_t^{(m)})\|^2 \\
& + \Big[\eta_\theta + 2c_{t+1}\eta_\theta + 9\big(\tfrac{L}{2}\eta_\theta^2 + c_{t+1}\eta_\theta^2\big)\Big]\frac{C_1}{M} \\
& + 9L_1\big(\tfrac{L}{2}\eta_\theta^2 + c_{t+1}\eta_\theta^2\big)\mathbb{E}\|\widetilde{\omega}^{(m)} - \omega^*(\widetilde{\theta}^{(m)})\|^2 \\
& + \Big[6L_1\big(\tfrac{L}{2}\eta_\theta^2 + c_{t+1}\eta_\theta^2\big) + \eta_\theta L_1 + \eta_\theta L_1 c_{t+1}\Big]\mathbb{E}\|\omega_t^{(m)} - \omega^*(\theta_t^{(m)})\|^2 \\
& - \Big[6L_1\big(\tfrac{L}{2}\eta_\theta^2 + c_{t+1}\eta_\theta^2\big) + \eta_\theta L_1 + \eta_\theta L_1 c_{t+1}\Big] \cdot \frac{4}{\lambda_C}\Big[12L_5^2\eta_\omega + \big(\tfrac{9}{\lambda_C} + 2L_3^2\big)9L_2^2\frac{\eta_\theta^2}{\eta_\omega^2}\Big]\mathbb{E}\|\theta_t^{(m)} - \widetilde{\theta}^{(m)}\|^2 \\
& + \Big[6L_1\big(\tfrac{L}{2}\eta_\theta^2 + c_{t+1}\eta_\theta^2\big) + \eta_\theta L_1 + \eta_\theta L_1 c_{t+1}\Big] \cdot \frac{4}{\lambda_C}\Big[12L_5^2\eta_\omega + \big(\tfrac{9}{\lambda_C} + 2L_3^2\big)9L_2^2\frac{\eta_\theta^2}{\eta_\omega^2}\Big]\mathbb{E}\|\theta_t^{(m)} - \widetilde{\theta}^{(m)}\|^2.
\end{aligned}
$$

Then, we define $R_t^m := J(\theta_t^{(m)}) + c_t\|\theta_t^{(m)} - \widetilde{\theta}^{(m)}\|^2$ with $c_t$ being specified via the following recursion.

$$
\begin{aligned}
c_t = & c_{t+1}(\eta_\theta \beta_t + 1 + 2\eta_\theta) + 9L_1\big(\tfrac{L}{2}\eta_\theta^2 + c_{t+1}\eta_\theta^2\big) \\
& + \Big[6L_1\big(\tfrac{L}{2}\eta_\theta^2 + c_{t+1}\eta_\theta^2\big) + \eta_\theta L_1 + \eta_\theta L_1 c_{t+1}\Big] \cdot \frac{4}{\lambda_C}\Big[12L_5^2\eta_\omega + \big(\tfrac{9}{\lambda_C} + 2L_3^2\big)9L_2^2\frac{\eta_\theta^2}{\eta_\omega^2}\Big]. \quad (11)
\end{aligned}
$$

Based on this definition, the previous inequality reduces to

$$
\begin{aligned}
\mathbb{E}[R_{t+1}^m] \leq \mathbb{E}[R_t^m] & + \Big[-\tfrac{1}{2}\eta_\theta + c_{t+1}\tfrac{\eta_\theta}{\beta_t} + 9\big(\tfrac{L}{2}\eta_\theta^2 + c_{t+1}\eta_\theta^2\big)\Big]\mathbb{E}\|\nabla J(\theta_t^{(m)})\|^2 \\
& + \Big[\eta_\theta + 2c_{t+1}\eta_\theta + 9\big(\tfrac{L}{2}\eta_\theta^2 + c_{t+1}\eta_\theta^2\big)\Big]\frac{C_1}{M} \\
& + 9L_1\big(\tfrac{L}{2}\eta_\theta^2 + c_{t+1}\eta_\theta^2\big)\mathbb{E}\|\widetilde{\omega}^{(m)} - \omega^*(\widetilde{\theta}^{(m)})\|^2 \\
& + \Big[6L_1\big(\tfrac{L}{2}\eta_\theta^2 + c_{t+1}\eta_\theta^2\big) + \eta_\theta L_1 + \eta_\theta L_1 c_{t+1}\Big]\mathbb{E}\|\omega_t^{(m)} - \omega^*(\theta_t^{(m)})\|^2 \\
& - \Big[6L_1\big(\tfrac{L}{2}\eta_\theta^2 + c_{t+1}\eta_\theta^2\big) + \eta_\theta L_1 + \eta_\theta L_1 c_{t+1}\Big] \cdot \frac{4}{\lambda_C}\Big[12L_5^2\eta_\omega + \big(\tfrac{9}{\lambda_C} + 2L_3^2\big)9L_2^2\frac{\eta_\theta^2}{\eta_\omega^2}\Big]\mathbb{E}\|\theta_t^{(m)} - \widetilde{\theta}^{(m)}\|^2.
\end{aligned}
$$
$$(12)$$

Assume that $c_t \leq \widehat{c}$ for some universal constant $\widehat{c} > 0$ (we will formally prove it later). Then, we sum the above inequality over one epoch and obtain that

$$
\begin{aligned}
& \Big[\tfrac{1}{2}\eta_\theta - \widehat{c}\tfrac{\eta_\theta}{\beta_t} - 9\big(\tfrac{L}{2}\eta_\theta^2 + \widehat{c}\eta_\theta^2\big)\Big] \sum_{t=0}^{M-1} \mathbb{E}\|\nabla J(\theta_t^{(m)})\|^2 \\
& \leq \mathbb{E}[R_0^m] - \mathbb{E}[R_M^m] + \Big[\eta_\theta + 2\widehat{c}\eta_\theta + 9\big(\tfrac{L}{2}\eta_\theta^2 + \widehat{c}\eta_\theta^2\big)\Big]C_1 + 9L_1\big(\tfrac{L}{2}\eta_\theta^2 + \widehat{c}\eta_\theta^2\big)M\mathbb{E}\|\widetilde{\omega}^{(m)} - \omega^*(\widetilde{\theta}^{(m)})\|^2 \\
& \quad + \Big[6L_1\big(\tfrac{L}{2}\eta_\theta^2 + \widehat{c}\eta_\theta^2\big) + \eta_\theta L_1 + \eta_\theta L_1\widehat{c}\Big] \sum_{t=0}^{M-1} \mathbb{E}\|\omega_t^{(m)} - \omega^*(\theta_t^{(m)})\|^2 \\
& \quad - \Big[6L_1\big(\tfrac{L}{2}\eta_\theta^2 + \widehat{c}\eta_\theta^2\big) + \eta_\theta L_1 + \eta_\theta L_1\widehat{c}\Big] \cdot \frac{4}{\lambda_C}\Big[12L_5^2\eta_\omega + \big(\tfrac{9}{\lambda_C} + 2L_3^2\big)9L_2^2\frac{\eta_\theta^2}{\eta_\omega^2}\Big] \sum_{t=0}^{M-1} \mathbb{E}\|\theta_t^{(m)} - \widetilde{\theta}^{(m)}\|^2.
\end{aligned}
$$
$$(13)$$

By Lemma D.7, we have that

$$\sum_{t=0}^{M-1} \mathbb{E}\|\omega_t^{(m)} - \omega^*(\theta_t^{(m)})\|^2$$

$$\leq \frac{4}{\lambda_C}\Big[\frac{1}{\eta_\omega} + M\Big[\big(\frac{9}{\lambda_C} + 2L_3^2\big)9L_1^2\frac{\eta_\theta^2}{\eta_\omega^2} + 18L_4^2\eta_\omega\Big]\Big]\mathbb{E}\|\widetilde{\omega}^{(m)} - \omega^*(\widetilde{\theta}^{(m)})\|^2$$

$$+ \frac{4}{\lambda_C}\big(\frac{9}{\lambda_C} + 2L_3^2\big)\frac{\eta_\theta^2}{\eta_\omega^2}\cdot 9C_1 + \frac{4}{\lambda_C}\eta_\omega\cdot 12C_2 + \big(\frac{9}{\lambda_C} + 2L_3^2\big)\frac{\eta_\theta^2}{\eta_\omega^2}\frac{36}{\lambda_C}\sum_{t=0}^{M-1}\mathbb{E}\|\nabla J(\theta_t^{(m)})\|^2$$

$$+ \frac{4}{\lambda_C}\Big[12L_5^2\eta_\omega + \big(\frac{9}{\lambda_C} + 2L_3^2\big)9L_2^2\frac{\eta_\theta^2}{\eta_\omega^2}\Big]\sum_{t=0}^{M-1}\mathbb{E}\|\theta_t^{(m)} - \widetilde{\theta}^{(m)}\|^2$$

$$+ \frac{8}{\lambda_C}(C_3 + C_4).$$

For simplicity, we define $D := 6L_1\big(\frac{L}{2} + \widehat{c}\big) + L_1 + L_1\widehat{c}$. Substituting the above bound into the previous inequality and simplifying, we obtain that

$$\Big[\frac{1}{2}\eta_\theta - \widehat{c}\frac{\eta_\theta}{\beta_t} - 9\big(\frac{L}{2}\eta_\theta^2 + \widehat{c}\eta_\theta^2\big)\Big]\sum_{t=0}^{M-1}\mathbb{E}\|\nabla J(\theta_t^{(m)})\|^2$$

$$\leq \mathbb{E}R_0^m - \mathbb{E}R_M^m + \Big[\eta_\theta + 2\widehat{c}\eta_\theta + 9\big(\frac{L}{2}\eta_\theta^2 + \widehat{c}\eta_\theta^2\big)\Big]C_1 + 9L_1\big(\frac{L}{2}\eta_\theta^2 + \widehat{c}\eta_\theta^2\big)M\mathbb{E}\|\widetilde{\omega}^{(m)} - \omega^*(\widetilde{\theta}^{(m)})\|^2$$

$$+ D\eta_\theta\Big[\frac{4}{\lambda_C}\Big[\frac{1}{\eta_\omega} + M\Big[\big(\frac{9}{\lambda_C} + 2L_3^2\big)9L_1^2\frac{\eta_\theta^2}{\eta_\omega^2} + 18L_4^2\eta_\omega\Big]\Big]\mathbb{E}\|\widetilde{\omega}^{(m)} - \omega^*(\widetilde{\theta}^{(m)})\|^2$$

$$+ \frac{4}{\lambda_C}\big(\frac{9}{\lambda_C} + 2L_3^2\big)\frac{\eta_\theta^2}{\eta_\omega^2}\cdot 9C_1 + \frac{4}{\lambda_C}\eta_\omega\cdot 12C_2 + \big(\frac{9}{\lambda_C} + 2L_3^2\big)\frac{\eta_\theta^2}{\eta_\omega^2}\frac{36}{\lambda_C}\sum_{t=0}^{M-1}\mathbb{E}\|\nabla J(\theta_t^{(m)})\|^2 + \frac{8}{\lambda_C}(C_3 + C_4).\Big]$$

One can see that the above bound is independent of $\sum_{t=0}^{M-1}\mathbb{E}\|\theta_t^{(m+1)} - \widetilde{\theta}^{(m)}\|^2$, and this is what we desire. After simplification, the above inequality further implies that

$$\Big[\frac{1}{2}\eta_\theta - \widehat{c}\frac{\eta_\theta}{\beta_t} - 9\big(\frac{L}{2}\eta_\theta^2 + \widehat{c}\eta_\theta^2\big) - D\big(\frac{9}{\lambda_C} + 2L_3^2\big)\frac{36}{\lambda_C}\frac{\eta_\theta^3}{\eta_\omega^2}\Big]\sum_{t=0}^{M-1}\mathbb{E}\|\nabla J(\theta_t^{(m)})\|^2$$

$$\leq \mathbb{E}[R_0^m] - \mathbb{E}[R_M^m]$$

$$+ \Big[\eta_\theta + 2\widehat{c}\eta_\theta + 9\big(\frac{L}{2}\eta_\theta^2 + \widehat{c}\eta_\theta^2\big)\Big]C_1 + \frac{8}{\lambda_C}(C_3 + C_4)D\eta_\theta + D\eta_\theta\Big[\frac{4}{\lambda_C}\big(\frac{9}{\lambda_C} + 2L_3^2\big)\frac{\eta_\theta^2}{\eta_\omega^2}\cdot 9C_1 + \frac{4}{\lambda_C}\eta_\omega\cdot 12C_2\Big]$$

$$+ \Big[9L_1\big(\frac{L}{2}\eta_\theta^2 + \widehat{c}\eta_\theta^2\big)M + D\eta_\theta\Big[\frac{4}{\lambda_C}\Big[\frac{1}{\eta_\omega} + M\Big[\big(\frac{9}{\lambda_C} + 2L_3^2\big)9L_1^2\frac{\eta_\theta^2}{\eta_\omega^2} + 18L_4^2\eta_\omega\Big]\Big]\Big]\mathbb{E}\|\widetilde{\omega}^{(m)} - \omega^*(\widetilde{\theta}^{(m)})\|^2.$$

$$(14)$$

**Choose optimal learning rates:** Here, we provide the omitted proof of our earlier claim made after eq. (12), that is, the upper bound of $\{c_t\}$ is a small constant. We first present the following fundamental simple lemma, and the proof is omitted.

**Lemma B.1.** *Let $\{c_i\}_{i=0,\dots,M}$ be a finite sequence with $c_M = 0$ and satisfies the following relation for certain $\mathrm{a} > 1$:*

$$c_t \leq \mathrm{a}\cdot c_{t+1} + \mathrm{b}.$$

*Then, $\{c_i\}_{i=0,\dots,M}$ is a deceasing sequence and*

$$c_0 \leq \mathrm{ab}\cdot\frac{\mathrm{a}^M - 1}{\mathrm{a} - 1}.$$

Next, we derive the upper bound $\widehat{c}$ of $c_t$. Set $\beta_t = 1$ for all $t$. Then we have that

$$c_t \leq c_{t+1}(\eta_\theta + 1 + 2\eta_\theta) + 9L_1\left(\frac{L}{2}\eta_\theta^2 + c_{t+1}\eta_\theta^2\right)$$

$$+ \left[6L_1\left(\frac{L}{2}\eta_\theta^2 + c_{t+1}\eta_\theta^2\right) + \eta_\theta L_1 + \eta_\theta L_1 c_{t+1}\right] \cdot \frac{4}{\lambda_C}\left[12L_5^2\eta_\omega + \left(\frac{9}{\lambda_C} + 2L_3^2\right)9L_2^2\frac{\eta_\theta^2}{\eta_\omega^2}\right]$$

$$:= \mathrm{a} \cdot c_{t+1} + \mathrm{b}$$

where

$$\mathrm{a} = 1 + (3 + 16L_1)\eta_\theta$$

and

$$\mathrm{b} = \frac{15}{2}L_1 L\eta_\theta^2 + L_1\eta_\theta \cdot \frac{4}{\lambda_C}\left[12L_5^2\eta_\omega + \left(\frac{9}{\lambda_C} + 2L_3^2\right)9L_2^2\frac{\eta_\theta^2}{\eta_\omega^2}\right].$$

Note that here we require

$$\frac{4}{\lambda_C}\left[12L_5^2\eta_\omega + \left(\frac{9}{\lambda_C} + 2L_3^2\right)9L_2^2\frac{\eta_\theta^2}{\eta_\omega^2}\right] \leq 1 \tag{15}$$

and

$$\max\{\eta_\omega, \eta_\theta\} \leq 1. \tag{16}$$

Moreover, let

$$(3 + 16L_1)\eta_\theta \leq \frac{1}{M}. \tag{17}$$

Based on the above conditions, we obtain that

$$c_0 \leq \left[\frac{15}{2}L_1 L\eta_\theta + \frac{4L_1}{\lambda_C}\left[12L_5^2\eta_\omega + \left(\frac{9}{\lambda_C} + 2L_3^2\right)9L_2^2\frac{\eta_\theta^2}{\eta_\omega^2}\right]\right] \cdot \frac{4 + 16L_1}{3 + 16L_1} \cdot \left[(1 + (3 + 16L_1)\eta_\theta)^M - 1\right]$$

$$\leq \left[\frac{15}{2}L_1 L\eta_\theta + \frac{4L_1}{\lambda_C}\left[12L_5^2\eta_\omega + \left(\frac{9}{\lambda_C} + 2L_3^2\right)9L_2^2\frac{\eta_\theta^2}{\eta_\omega^2}\right]\right] \cdot \frac{4 + 16L_1}{3 + 16L_1} \cdot (e - 1).$$

Lastly, we choose

$$\left[\frac{15}{2}L_1 L\eta_\theta + \frac{4L_1}{\lambda_C}\left[12L_5^2\eta_\omega + \left(\frac{9}{\lambda_C} + 2L_3^2\right)9L_2^2\frac{\eta_\theta^2}{\eta_\omega^2}\right]\right] \cdot \frac{4 + 16L_1}{3 + 16L_1} \cdot (e - 1) \leq \frac{1}{8}. \tag{18}$$

Therefore $c_0 \leq \frac{1}{8}$. Since $\{c_t\}_t$ is decreasing, we obtain that $\widehat{c} = \frac{1}{8}$. Now, substituting $\beta_t = 1$ and $\widehat{c} = \frac{1}{8}$ into the coefficient of the term $\sum_{t=0}^{M-1} \mathbb{E}\|\nabla J(\theta_t^{(m)})\|^2$ in eq. (14), the coefficient reduces to the following, and we choose an appropriate $(\eta_\theta, \eta_\omega)$ such that the coefficient is greater than $\frac{1}{4}\eta_\theta$.

$$\frac{3}{8}\eta_\theta - 9\left(\frac{L}{2}\eta_\theta^2 + \widehat{c}\eta_\theta^2\right) - D\left(\frac{9}{\lambda_C} + 2L_3^2\right)\frac{36}{\lambda_C}\frac{\eta_\theta^3}{\eta_\omega^2} \geq \frac{1}{4}\eta_\theta. \tag{19}$$

**Deriving the final bound:** Exploiting the above conditions on the learning rates, eq. (14) further implies that

$$\frac{1}{4}\eta_\theta \sum_{t=0}^{M-1} \mathbb{E}\|\nabla J(\theta_t^{(m)})\|^2$$

$$\leq \mathbb{E}[J(\widetilde{\theta}^{(m)})] - \mathbb{E}[J(\widetilde{\theta}^{(m+1)})]$$

$$+ \left[\eta_\theta + 2\widehat{c}\eta_\theta + 9\left(\frac{L}{2}\eta_\theta^2 + \widehat{c}\eta_\theta^2\right)\right]C_1 + \frac{8}{\lambda_C}(C_3 + C_4)D\eta_\theta + D\eta_\theta\left[\frac{4}{\lambda_C}\left(\frac{9}{\lambda_C} + 2L_3^2\right)\frac{\eta_\theta^2}{\eta_\omega^2} \cdot 9C_1 + \frac{4}{\lambda_C}\eta_\omega \cdot 12C_2\right]$$

$$+ \left[9L_1\left(\frac{L}{2}\eta_\theta^2 + \widehat{c}\eta_\theta^2\right)M + D\eta_\theta\left[\frac{4}{\lambda_C}\left[\frac{1}{\eta_\omega} + M\left[\left(\frac{9}{\lambda_C} + 2L_3^2\right)9L_1^2\frac{\eta_\theta^2}{\eta_\omega^2} + 18L_4^2\eta_\omega\right]\right]\right]\mathbb{E}\|\widetilde{\omega}^{(m)} - \omega^*(\widetilde{\theta}^{(m)})\|^2. \tag{20}$$

On the other hand, by Lemma D.8 we have that

$$\mathbb{E}\|\widetilde{\omega}^{(m)} - \omega^*(\widetilde{\theta}^{(m)})\|^2 \le (1 - \frac{1}{2}\lambda_C\eta_\omega)^{mM}\mathbb{E}\|\widetilde{\omega}^{(0)} - \omega^*(\widetilde{\theta}^{(0)})\|^2$$
$$+ \frac{4}{\lambda_C}(C_3 + C_4)\frac{1}{M} + \frac{4}{\lambda_C}H^2\eta_\omega + \frac{2}{\lambda_C}(2L_3^2G^2 + \frac{9}{\lambda_C}G^2)\frac{\eta_\theta^2}{\eta_\omega^2}.$$

Substituting the above bound into eq. (20) and summing over $m$, we obtain that

$$\frac{1}{4}\eta_\theta\frac{1}{TM}\sum_{m=1}^{T}\sum_{t=0}^{M-1}\mathbb{E}\|\nabla J(\theta_t^{(m)})\|^2$$

$$\le \frac{1}{TM}\mathbb{E}[J(\widetilde{\theta}^{(0)})]$$

$$+ \frac{1}{M}\Big\{\Big[\eta_\theta + 2\widehat{c}\eta_\theta + 9(\frac{L}{2}\eta_\theta^2 + \widehat{c}\eta_\theta^2)\Big]C_1 + \frac{8}{\lambda_C}(C_3 + C_4)D\eta_\theta + D\eta_\theta\Big[\frac{4}{\lambda_C}(\frac{9}{\lambda_C} + 2L_3^2)\frac{\eta_\theta^2}{\eta_\omega^2}\cdot 9C_1 + \frac{4}{\lambda_C}\eta_\omega\cdot 12C_2\Big]\Big\}$$

$$+ \frac{1}{TM}\Big[9L_1(\frac{L}{2}\eta_\theta^2 + \widehat{c}\eta_\theta^2)M$$

$$+ D\eta_\theta\Big[\frac{4}{\lambda_C}\Big[\frac{1}{\eta_\omega} + M\Big[(\frac{9}{\lambda_C} + 2L_3^2)9L_1^2\frac{\eta_\theta^2}{\eta_\omega^2} + 18L_4^2\eta_\omega\Big]\Big]\Big] \cdot \mathbb{E}\|\widetilde{\omega}^{(0)} - \omega^*(\widetilde{\theta}^{(0)})\|^2 \cdot \frac{1}{1 - (1 - \frac{1}{2}\lambda_C\eta_\omega)^M}$$

$$+ \frac{1}{M}\Big[9L_1(\frac{L}{2}\eta_\theta^2 + \widehat{c}\eta_\theta^2)M + D\eta_\theta\Big[\frac{4}{\lambda_C}\Big[\frac{1}{\eta_\omega} + M\Big[(\frac{9}{\lambda_C} + 2L_3^2)9L_1^2\frac{\eta_\theta^2}{\eta_\omega^2} + 18L_4^2\eta_\omega\Big]\Big]\Big]$$

$$\cdot \Big[\frac{4}{\lambda_C}(C_3 + C_4)\frac{1}{M} + \frac{4}{\lambda_C}H^2\eta_\omega + \frac{2}{\lambda_C}(2L_3^2G^2 + \frac{9}{\lambda_C}G^2)\frac{\eta_\theta^2}{\eta_\omega^2}\Big]\Big].$$

Rearranging the above inequality, we obtain the following final bound, where $\xi, \zeta$ are random indexes that are sampled from $\{0, ..., M-1\}$ and $\{1, ..., T\}$ uniformly at random, respectively.

$$\mathbb{E}\|\nabla J(\theta_\xi^{(\zeta)})\|^2$$

$$\le \frac{1}{\eta_\theta TM}\cdot 4\mathbb{E}[J(\widetilde{\theta}^{(0)})]$$

$$+ \frac{4}{M}\Big\{\Big[1 + 2\widehat{c} + 9(\frac{L}{2}\eta_\theta + \widehat{c}\eta_\theta)\Big]C_1 + \frac{8}{\lambda_C}(C_3 + C_4)D + D\Big[\frac{4}{\lambda_C}(\frac{9}{\lambda_C} + 2L_3^2)\frac{\eta_\theta^2}{\eta_\omega^2}\cdot 9C_1 + \frac{4}{\lambda_C}\eta_\omega\cdot 12C_2\Big]\Big\}$$

$$+ \frac{1}{TM}\Big[9L_1(\frac{L}{2}\eta_\theta + \widehat{c}\eta_\theta)M + D\Big[\frac{4}{\lambda_C}\Big[\frac{1}{\eta_\omega} + M\Big[(\frac{9}{\lambda_C} + 2L_3^2)9L_1^2\frac{\eta_\theta^2}{\eta_\omega^2} + 18L_4^2\eta_\omega\Big]\Big]\Big]\Big]$$

$$\cdot \mathbb{E}\|\widetilde{\omega}^{(0)} - \omega^*(\widetilde{\theta}^{(0)})\|^2 \cdot \frac{1}{1 - (1 - \frac{1}{2}\lambda_C\eta_\omega)^M}$$

$$+ \Big[9L_1(\frac{L}{2}\eta_\theta + \widehat{c}\eta_\theta) + D\Big[\frac{4}{\lambda_C}\Big[\frac{1}{\eta_\omega M} + \Big[(\frac{9}{\lambda_C} + 2L_3^2)9L_1^2\frac{\eta_\theta^2}{\eta_\omega^2} + 18L_4^2\eta_\omega\Big]\Big]\Big]\Big]$$

$$\cdot \Big[\frac{4}{\lambda_C}(C_3 + C_4)\frac{1}{M} + \frac{4}{\lambda_C}H^2\eta_\omega + \frac{2}{\lambda_C}(2L_3^2G^2 + \frac{9}{\lambda_C}G^2)\frac{\eta_\theta^2}{\eta_\omega^2}\Big]\Big].$$

Next, we simplify the above inequality into an asymptotic form. Note that the first term is in the order of $\mathcal{O}(\frac{1}{\eta_\theta TM})$. The second term is of order $\mathcal{O}(\frac{1}{M})$. The third term is of order $\mathcal{O}(\frac{1}{\eta_\omega TM} + \frac{1}{T}(\eta_\omega + \frac{\eta_\theta^2}{\eta_\omega^2}))$, and the last term is the product of a term of order $\mathcal{O}(\frac{\eta_\theta^2}{\eta_\omega^2} + \eta_\omega + \frac{1}{\eta_\omega M})$ and another term of order $\mathcal{O}(\frac{1}{M} + \frac{\eta_\theta^2}{\eta_\omega^2} + \eta_\omega)$, which leads to the overall order $\mathcal{O}((\frac{\eta_\theta^2}{\eta_\omega^2} + \eta_\omega)^2 + \frac{1}{M})$. Combining these asymptotic orders together, we obtain the following asymptotic convergence rate result.

$$\mathbb{E}\|\nabla J(\theta_\xi^{(\zeta)})\|^2 = \mathcal{O}\Big(\frac{1}{\eta_\theta TM} + (\frac{\eta_\theta^2}{\eta_\omega^2} + \eta_\omega)^2 + \frac{1}{M} + \frac{1}{T}(\eta_\omega + \frac{\eta_\theta^2}{\eta_\omega^2})\Big).$$

## C    PROOF OF COROLLARY 4.6

Regarding the convergence rate result of Theorem 4.5, we choose the optimized learning rates such that $\eta_\theta = \mathcal{O}(\eta_\omega^{3/2})$, and we obtain that

$$\mathbb{E}\|\nabla J(\theta_\xi^{(\zeta)})\|^2 = \mathcal{O}\Big(\frac{1}{\eta_\theta T M} + \eta_\omega^2 + \frac{1}{M} + \frac{\eta_\omega}{T}\Big).$$

Then, we set $\eta_\theta = \mathcal{O}(\frac{1}{M})$ such that eq. (17) is satisfied, and moreover $\eta_\omega = \mathcal{O}(\frac{1}{M^{2/3}})$. Under this learning rate setting, the learning rate conditions in eq. (15), eq. (16), eq. (18), eq. (19) are all satisfied for a sufficiently large constant-level $M$. Then, the overall convergence rate further becomes

$$\mathbb{E}\|\nabla J(\theta_\xi^{(\zeta)})\|^2 = \mathcal{O}\Big(\frac{1}{T} + \frac{1}{M}\Big). \tag{21}$$

By choosing $T, M = \mathcal{O}(\epsilon^{-1})$, we conclude that the sample complexity for achieving $\mathbb{E}\|\nabla J(\theta_\xi^{(\zeta)})\|^2 \le \epsilon$ is in the order of $TM = \mathcal{O}(\epsilon^{-2})$.

## D    TECHNICAL LEMMAS

In this section, we present all the technical lemmas that are used in the proof of the main theorem.

**Bounding $\mathbb{E}\|g_t^{(m)}\|^2$ and $\mathbb{E}\|h_t^{(m)}\|^2$:**

**Lemma D.1.** *Under the same assumptions as those of Theorem 4.5, the square norm of the one-step update of $\theta_t^{(m)}$ in Algorithm 1 is bounded as*

$$\mathbb{E}\|g_t^{(m)}\|^2 \le 6L_1^2\mathbb{E}\|\omega_t^{(m)} - \omega^*(\theta_t^{(m)})\|^2 + 9L_1^2\mathbb{E}\|\widetilde{\omega}^{(m)} - \omega^*(\widetilde{\theta}^{(m)})\|^2 + 9L_2^2\mathbb{E}\|\theta_t^{(m)} - \widetilde{\theta}^{(m)}\|^2$$
$$+ \frac{1}{M}9C_1 + 9\mathbb{E}\|\nabla J(\theta_t^{(m)})\|^2$$

*where the constant $C_1$ is specified in Lemma D.3.*

*Proof.* For convenience, define

$$\mathscr{T}_t^{(m)} := G_t^{(m)}(\theta_t^{(m)}, \omega_t^{(m)}) - G_t^{(m)}(\widetilde{\theta}^{(m)}, \widetilde{\omega}^{(m)}),$$

and

$$\mathscr{S}_t^{(m)} := G_t^{(m)}(\theta_t^{(m)}, \omega^*(\theta_t^{(m)})) - G_t^{(m)}(\widetilde{\theta}^{(m)}, \omega^*(\widetilde{\theta}^{(m)})).$$

Then, we obtain that

$$\begin{aligned}
\|g_t^{(m)}\|^2 &= \|G_t^{(m)}(\theta_t^{(m)}, \omega_t^{(m)}) - G_t^{(m)}(\widetilde{\theta}^{(m)}, \widetilde{\omega}^{(m)}) + \widetilde{G}^{(m)}\|^2 \\
&= \|\mathscr{T}_t^{(m)} + \widetilde{G}^{(m)} - \widetilde{G}^{(m)}(\widetilde{\theta}^{(m)}, \omega^*(\widetilde{\theta}^{(m)})) + \widetilde{G}^{(m)}(\widetilde{\theta}^{(m)}, \omega^*(\widetilde{\theta}^{(m)})) \\
&\quad - \mathscr{S}_t^{(m)} + \mathscr{S}_t^{(m)}\|^2 \\
&\le 3\|\mathscr{T}_t^{(m)} - \mathscr{S}_t^{(m)}\|^2 + 3\|\widetilde{G}^{(m)} - \widetilde{G}^{(m)}(\widetilde{\theta}^{(m)}, \omega^*(\widetilde{\theta}^{(m)}))\|^2 \\
&\quad + 3\|\mathscr{S}_t^{(m)} + \widetilde{G}^{(m)}(\widetilde{\theta}^{(m)}, \omega^*(\widetilde{\theta}^{(m)}))\|^2 \\
&\le 6L_1^2\|\omega_t^{(m)} - \omega^*(\theta_t^{(m)})\|^2 + 9L_1^2\|\widetilde{\omega}^{(m)} - \omega^*(\widetilde{\theta}^{(m)})\|^2 \\
&\quad + 3\|\mathscr{S}_t^{(m)} + \widetilde{G}^{(m)}(\widetilde{\theta}^{(m)}, \omega^*(\widetilde{\theta}^{(m)}))\|^2,
\end{aligned}$$

where $\widetilde{G}^{(m)}(\widetilde{\theta}^{(m)}, \omega^*(\widetilde{\theta}^{(m)}))$ is obtained by substituting the arguments $\widetilde{\theta}^{(m)}, \omega^*(\widetilde{\theta}^{(m)})$ into the definition in eq. (4). Moreover, we have that

$$\begin{aligned}
&\|\mathscr{S}_t^{(m)} + \widetilde{G}^{(m)}(\widetilde{\theta}^{(m)}, \omega^*(\widetilde{\theta}^{(m)}))\|^2 \\
&= \|\mathscr{S}_t^{(m)} + \widetilde{G}^{(m)}(\widetilde{\theta}^{(m)}, \omega^*(\widetilde{\theta}^{(m)})) - \nabla J(\theta_t^{(m)}) + \nabla J(\theta_t^{(m)})\|^2 \\
&\le 3\|\mathscr{S}_t^{(m)} - \mathbb{E}_{m,t}\mathscr{S}_t^{(m)}\|^2 + 3\|\widetilde{G}^{(m)}(\theta_t^{(m)}, \omega^*(\theta_t^{(m)})) - \nabla J(\theta_t^{(m)})\|^2 \\
&\quad + 3\|\nabla J(\theta_t^{(m)})\|^2,
\end{aligned}$$

which further implies that

$$
\begin{aligned}
\mathbb{E}\|\mathscr{S}_t^{(m)} &+ \widetilde{G}^{(m)}(\widetilde{\theta}^{(m)}, \omega^*(\widetilde{\theta}^{(m)}))\|^2 \\
&\leq 3\mathbb{E}\|\mathscr{S}_t^{(m)}\|^2 + 3\mathbb{E}\|\widetilde{G}^{(m)}(\theta_t^{(m)}, \omega^*(\theta_t^{(m)})) - \nabla J(\theta_t^{(m)})\|^2 + 3\mathbb{E}\|\nabla J(\theta_t^{(m)})\|^2 \\
&\leq 3L_2^2 \mathbb{E}\|\theta_t^{(m)} - \widetilde{\theta}^{(m)}\|^2 + \frac{1}{M} \cdot 3C_1 + 3\mathbb{E}\|\nabla J(\theta_t^{(m)})\|^2.
\end{aligned}
$$

Combining all the above bounds, we finally obtain that

$$
\begin{aligned}
\mathbb{E}\|g_t^{(m)}\|^2 \leq 6L_1^2 \mathbb{E}\|\omega_t^{(m)} &- \omega^*(\theta_t^{(m)})\|^2 + 9L_1^2 \mathbb{E}\|\widetilde{\omega}^{(m)} - \omega^*(\widetilde{\theta}^{(m)})\|^2 \\
&+ 9L_2^2 \mathbb{E}\|\theta_t^{(m)} - \widetilde{\theta}^{(m)}\|^2 + \frac{1}{M} \cdot 9C_1 + 9\mathbb{E}\|\nabla J(\theta_t^{(m)})\|^2.
\end{aligned}
$$

$\square$

**Lemma D.2.** *Under the same assumptions as those of Theorem 4.5, we have that*

$$
\|h_t^{(m)}\|^2 \leq 6L_4^2 \|\omega_t^{(m)} - \omega^*(\theta_t^{(m)})\|^2 + 9L_4^2 \|\widetilde{\omega}^{(m)} - \omega^*(\widetilde{\theta}^{(m)})\|^2 + 6L_5^2 \|\theta_t^{(m)} - \widetilde{\theta}^{(m)}\|^2 + \frac{6C_2}{M}.
$$

*Proof.* For convenience, define

$$
\mathscr{V}_t^{(m)} := H_t^{(m)}(\theta_t^{(m)}, \omega_t^{(m)}) - H_t^{(m)}(\widetilde{\theta}^{(m)}, \widetilde{\omega}^{(m)}),
$$

and

$$
\mathscr{U}_t^{(m)} := H_t^{(m)}(\theta_t^{(m)}, \omega^*(\theta_t^{(m)})) - H_t^{(m)}(\widetilde{\theta}^{(m)}, \omega^*(\widetilde{\theta}^{(m)})).
$$

Then, we obtain that

$$
\begin{aligned}
\|h_t^{(m)}\|^2 &= \|H_t^{(m)}(\theta_t^{(m)}, \omega_t^{(m)}) - H_t^{(m)}(\widetilde{\theta}^{(m)}, \widetilde{\omega}^{(m)}) + \widetilde{H}^{(m)}\|^2 \\
&= \|\mathscr{V}_t^{(m)} + \widetilde{H}^{(m)} - \widetilde{H}^{(m)}(\widetilde{\theta}^{(m)}, \omega^*(\widetilde{\theta}^{(m)})) + \widetilde{H}^{(m)}(\widetilde{\theta}^{(m)}, \omega^*(\widetilde{\theta}^{(m)})) - \mathscr{U}_t^{(m)} + \mathscr{U}_t^{(m)}\|^2 \\
&\leq 3\|\mathscr{V}_t^{(m)} - \mathscr{U}_t^{(m)}\|^2 + 3\|\widetilde{H}^{(m)} - \widetilde{H}^{(m)}(\widetilde{\theta}^{(m)}, \omega^*(\widetilde{\theta}^{(m)}))\|^2 \\
&\quad + 3\|\mathscr{U}_t^{(m)} + \widetilde{H}^{(m)}(\widetilde{\theta}^{(m)}, \omega^*(\widetilde{\theta}^{(m)}))\|^2 \\
&\leq 6L_4^2 \|\omega_t^{(m)} - \omega^*(\theta_t^{(m)})\|^2 + 9L_4^2 \|\widetilde{\omega}^{(m)} - \omega^*(\widetilde{\theta}^{(m)})\|^2 \\
&\quad + 3\|\mathscr{U}_t^{(m)} + \widetilde{H}^{(m)}(\widetilde{\theta}^{(m)}, \omega^*(\widetilde{\theta}^{(m)}))\|^2.
\end{aligned}
$$

Moreover, note that

$$
\begin{aligned}
\|\mathscr{U}_t^{(m)} + \widetilde{H}^{(m)}(\widetilde{\theta}^{(m)}, \omega^*(\widetilde{\theta}^{(m)}))\|^2 &\leq 2\|\mathscr{U}_t^{(m)}\|^2 + 2\|\widetilde{H}^{(m)}(\widetilde{\theta}^{(m)}, \omega^*(\widetilde{\theta}^{(m)}))\|^2 \\
&\leq 2L_5^2 \|\theta_t^{(m)} - \widetilde{\theta}^{(m)}\|^2 + \frac{2C_2}{M}.
\end{aligned}
$$

Combining the above bounds, we finally obtain that

$$
\begin{aligned}
\|h_t^{(m)}\|^2 \leq 6L_4^2 \|\omega_t^{(m)} &- \omega^*(\theta_t^{(m)})\|^2 + 9L_4^2 \|\widetilde{\omega}^{(m)} - \omega^*(\widetilde{\theta}^{(m)})\|^2 \\
&+ 6L_5^2 \|\theta_t^{(m)} - \widetilde{\theta}^{(m)}\|^2 + \frac{6C_2}{M}.
\end{aligned}
$$

$\square$

**Bounding pseudo-gradient variance:**

**Lemma D.3.** *Under the same assumptions as those of Theorem 4.5, we have that*

$$
\mathbb{E}\|\widetilde{G}^{(m)}(\theta_t^{(m)}, \omega^*(\theta_t^{(m)})) - \nabla J(\theta_t^{(m)})\|^2 \leq \frac{C_1}{M}.
$$

*Proof.* Note that the variance can be expanded as

$$
\mathbb{E}\|\widetilde{G}^{(m)}(\theta_t^{(m)},\omega^*(\theta_t^{(m)})) - \nabla J(\theta_t^{(m)})\|^2
$$

$$
=\frac{1}{M^2}\mathbb{E}\big[\sum_{s=0}^{M-1}\|G_s^{(m)}(\theta_t^{(m)},\omega^*(\theta_t^{(m)})) - \nabla J(\theta_t^{(m)})\|^2
$$

$$
+\sum_{i\neq j}\langle G_i^{(m)}(\theta_t^{(m)},\omega^*(\theta_t^{(m)})) - \nabla J(\theta_t^{(m)}), G_j^{(m)}(\theta_t^{(m)},\omega^*(\theta_t^{(m)})) - \nabla J(\theta_t^{(m)})\rangle\big]
$$

$$
\leq\frac{1}{M^2}\big[\sum_{s=0}^{M-1}(G + C_{\nabla J})^2 + \sum_{i\neq j}\Lambda\rho^{|i-j|}(G + C_{\nabla J})^2\big]
$$

$$
\leq\frac{1}{M}(1 + 2\Lambda\frac{\rho}{1-\rho})(G + C_{\nabla J})^2.
$$

Then, we define the constant $C_1 := (1 + 2\Lambda\frac{\rho}{1-\rho})(G + C_{\nabla J})^2$. $\qquad\square$

**Lemma D.4.** *Under the same assumptions as those of Theorem 4.5, we have that*

$$
\mathbb{E}\|\widetilde{H}^{(m)}(\widetilde{\theta}^{(m)},\omega^*(\widetilde{\theta}^{(m)}))\|^2 \leq \frac{C_2}{M}.
$$

*Proof.* Note that this second moment term can be expanded as

$$
\mathbb{E}\|\widetilde{H}^{(m)}(\widetilde{\theta}^{(m)},\omega^*(\widetilde{\theta}^{(m)}))\|^2 = \frac{1}{M^2}\sum_{i=0}^{M-1}\mathbb{E}\|H_i^{(m)}(\widetilde{\theta}^{(m)},\omega^*(\widetilde{\theta}^{(m)}))\|
$$

$$
+\frac{1}{M^2}\sum_{i\neq j}\mathbb{E}\langle H_i^{(m)}(\widetilde{\theta}^{(m)},\omega^*(\widetilde{\theta}^{(m)})), H_j^{(m)}(\widetilde{\theta}^{(m)},\omega^*(\widetilde{\theta}^{(m)}))\rangle
$$

$$
\leq\frac{H^2}{M} + \frac{1}{M^2}H^2\Lambda\sum_{i\neq j}\rho^{|i-j|}
$$

$$
\leq H^2(1 + \Lambda\frac{\rho}{1-\rho})\frac{1}{M}.
$$

Lastly, we define the constant $C_2 := H^2(1 + \Lambda\frac{\rho}{1-\rho})$. $\qquad\square$

**Bounding Markovian Noise:**

**Lemma D.5.** *Let the same assumptions as those of Theorem 4.5 hold and define*

$$
\varsigma_t^{(m)} := \langle\omega_t^{(m)} - \omega^*(\theta_t^{(m)}), (\phi_t^{(m)}(\phi_t^{(m)})^\top - C)(\omega_t^{(m)} - \omega^*(\theta_t^{(m)}))\rangle.
$$

*Then, it holds that*

$$
\mathbb{E}[\varsigma_t^{(m)}] \leq \frac{1}{8}\lambda_C\|\omega_t^{(m)} - \omega^*(\theta_t^{(m)})\|^2 + \frac{C_3}{M},
$$

*where $C_3 = \frac{8R^2}{\lambda_C}(1 + \frac{\rho\Lambda}{1-\rho})$.*

*Proof.* By definition of $\varsigma_t^{(m)}$, we obtain that

$$
\mathbb{E}\langle\omega_t^{(m)} - \omega^*(\theta_t^{(m)}), (\phi_t^{(m)}(\phi_t^{(m)})^\top - C)(\omega_t^{(m)} - \omega^*(\theta_t^{(m)}))\rangle
$$

$$
=\mathbb{E}\langle\omega_t^{(m)} - \omega^*(\theta_t^{(m)}), \mathbb{E}_{m,t-1}(\phi_t^{(m)}(\phi_t^{(m)})^\top - C)(\omega_t^{(m)} - \omega^*(\theta_t^{(m)}))\rangle
$$

$$
\leq\frac{1}{2}\cdot\frac{\lambda_C}{4}\mathbb{E}\|\omega_t^{(m)} - \omega^*(\theta_t^{(m)})\|^2 + \frac{1}{2}\cdot\frac{4R^2}{\lambda_C M^2}\mathbb{E}\|\sum_{i=0}^{M-1}(\phi_i^{(m)}(\phi_i^{(m)})^\top - C)\|^2.
$$

For the last term, note that

$$\mathbb{E}\|\sum_{i=0}^{M-1}\big(\phi_i^{(m)}(\phi_i^{(m)})^\top - C\big)\|^2$$

$$= \mathbb{E}\sum_{i=0}^{M-1}\|\phi_i^{(m)}(\phi_i^{(m)})^\top - C\|^2 + \mathbb{E}\sum_{i\neq j}\langle\phi_i^{(m)}(\phi_i^{(m)})^\top - C, \phi_j^{(m)}(\phi_j^{(m)})^\top - C\rangle$$

$$\leq 4M + 4\sum_{i\neq j}\Lambda\rho^{|i-j|}$$

$$\leq 4M + 4M\frac{\rho\Lambda}{1-\rho}.$$

Combining the above bounds, we finally obtain that

$$\mathbb{E}\langle\omega_t^{(m)} - \omega^*(\theta_t^{(m)}), \big(\phi_t^{(m)}(\phi_t^{(m)})^\top - C\big)\big(\omega_t^{(m)} - \omega^*(\theta_t^{(m)}))\rangle$$

$$\leq\frac{\lambda_C}{8}\mathbb{E}\|\omega_t^{(m)} - \omega^*(\theta_t^{(m)})\|^2 + \frac{8R^2}{\lambda_C}(1 + \frac{\rho\Lambda}{1-\rho})\frac{1}{M}.$$

We define $C_3 := \frac{8R^2}{\lambda_C}(1 + \frac{\rho\Lambda}{1-\rho})$. □

**Lemma D.6.** *Let the same assumptions as those of Theorem 4.5 hold and define*

$$\varkappa_t^{(m)} := \langle\omega_t^{(m)} - \omega^*(\theta_t^{(m)}), \big[(\phi_t^{(m)})^\top\omega^*(\theta_t^{(m)}) - \delta_{t+1}^{(m)}(\theta_t^{(m)})\big]\phi_t^{(m)}\rangle.$$

*Then, we obtain that*

$$\mathbb{E}\varkappa_t^{(m)} \leq \frac{1}{8}\lambda_C\|\omega_t^{(m)} - \omega^*(\theta_t^{(m)})\|^2 + \frac{C_4}{M}.$$

*Proof.* Similar to the proof of Lemma D.6, we have that

$$\mathbb{E}\langle\omega_t^{(m)} - \omega^*(\theta_t^{(m)}), \big[(\phi_t^{(m)})^\top\omega^*(\theta_t^{(m)}) - \delta_{t+1}^{(m)}(\theta_t^{(m)})\big]\phi_t^{(m)}\rangle$$

$$\leq\frac{1}{2}\cdot\frac{\lambda_C}{4}\mathbb{E}\|\omega_t^{(m)} - \omega^*(\theta_t^{(m)})\|^2$$

$$+ \frac{1}{2}\cdot\frac{4}{\lambda_C M^2}\mathbb{E}\|\sum_{i=0}^{M-1}\big([(\phi_i^{(m)})^\top\omega^*(\theta_t^{(m)}) - \delta_{i+1}^{(m)}(\theta_t^{(m)})]\phi_i^{(m)}\big)\|^2.$$

For the last term, we can bound it as

$$\mathbb{E}\|\sum_{i=0}^{M-1}\big([(\phi_i^{(m)})^\top\omega^*(\theta_t^{(m)}) - \delta_{i+1}^{(m)}(\theta_t^{(m)})]\phi_i^{(m)}\big)\|^2$$

$$\leq (R(2+\gamma) + r_{\max})^2 M + (R(2+\gamma) + r_{\max})^2\frac{\rho\Lambda}{1-\rho}M.$$

Combining all the above bounds, we finally obtain that

$$\mathbb{E}\langle\omega_t^{(m)} - \omega^*(\theta_t^{(m)}), \big[(\phi_t^{(m)})^\top\omega^*(\theta_t^{(m)}) - \delta_{t+1}^{(m)}(\theta_t^{(m)})\big]\phi_t^{(m)}\rangle$$

$$\leq\frac{\lambda_C}{8}\mathbb{E}\|\omega_t^{(m)} - \omega^*(\theta_t^{(m)})\|^2 + \frac{2}{\lambda_C}(R(2+\gamma) + r_{\max})^2(1 + \frac{\rho\Lambda}{1-\rho})\frac{1}{M}.$$

We then define $C_4 := \frac{2}{\lambda_C}(R(2+\gamma) + r_{\max})^2(1 + \frac{\rho\Lambda}{1-\rho})$. □

**Bounding Tracking Error:**

**Lemma D.7.** *Under the same assumptions as those of Theorem 4.5, the tracking error can be bounded as*

$$\sum_{t=0}^{M-1} \mathbb{E}\|\omega_t^{(m)} - \omega^*(\theta_t^{(m)})\|^2$$

$$\leq \frac{4}{\lambda_C}\Big[\frac{1}{\eta_\omega} + M\Big[\big(\frac{9}{\lambda_C} + 2L_3^2\big)9L_1^2\frac{\eta_\theta^2}{\eta_\omega^2} + 18L_4^2\eta_\omega\Big]\Big]\mathbb{E}\|\widetilde{\omega}^{(m)} - \omega^*(\widetilde{\theta}^{(m)})\|^2$$

$$+ \frac{4}{\lambda_C}\big(\frac{9}{\lambda_C} + 2L_3^2\big)\frac{\eta_\theta^2}{\eta_\omega^2}\cdot 9C_1 + \frac{4}{\lambda_C}\eta_\omega\cdot 12C_2 + \big(\frac{9}{\lambda_C} + 2L_3^2\big)\frac{\eta_\theta^2}{\eta_\omega^2}\frac{36}{\lambda_C}\sum_{t=0}^{M-1}\mathbb{E}\|\nabla J(\theta_t^{(m)})\|^2$$

$$+ \frac{4}{\lambda_C}\Big[12L_5^2\eta_\omega + \big(\frac{9}{\lambda_C} + 2L_3^2\big)9L_2^2\frac{\eta_\theta^2}{\eta_\omega^2}\Big]\sum_{t=0}^{M-1}\mathbb{E}\|\theta_t^{(m)} - \widetilde{\theta}^{(m)}\|^2$$

$$+ \frac{8}{\lambda_C}(C_3 + C_4).$$

*Proof.* Recall the one-step update at $\omega_{t+1}^{(m)}$:

$$\omega_{t+1}^{(m)} = \Pi_R\big(\omega_t^{(m)} - \eta_\omega h_t^{(m)}\big).$$

Then, we obtain the following upper bound of the tracking error $\|\omega_{t+1}^{(m)} - \omega^*(\theta_{t+1}^{(m)})\|^2$,

$$\|\omega_{t+1}^{(m)} - \omega^*(\theta_{t+1}^{(m)})\|^2 \leq \|\omega_t^{(m)} - \omega^*(\theta_t^{(m)}) - \eta_\omega h_t^{(m)} + \omega^*(\theta_t^{(m)}) - \omega^*(\theta_{t+1}^{(m)})\|^2$$

$$\leq \|\omega_t^{(m)} - \omega^*(\theta_t^{(m)})\|^2 - 2\eta_\omega\langle\omega_t^{(m)} - \omega^*(\theta_t^{(m)}), h_t^{(m)}\rangle$$

$$+ 2\langle\omega_t^{(m)} - \omega^*(\theta_t^{(m)}), \omega^*(\theta_t^{(m)}) - \omega^*(\theta_{t+1}^{(m)})\rangle$$

$$+ 2\eta_\omega^2\|h_t^{(m)}\|^2 + 2\|\omega^*(\theta_t^{(m)}) - \omega^*(\theta_{t+1}^{(m)})\|^2.$$

Substituting the bound of Lemma D.2 into the above bound, we obtain that

$$\|\omega_{t+1}^{(m)} - \omega^*(\theta_{t+1}^{(m)})\|^2$$

$$\leq \|\omega_t^{(m)} - \omega^*(\theta_t^{(m)})\|^2 - 2\eta_\omega\langle\omega_t^{(m)} - \omega^*(\theta_t^{(m)}), h_t^{(m)}\rangle$$

$$+ \lambda_C\eta_\omega\|\omega_t^{(m)} - \omega^*(\theta_t^{(m)})\|^2$$

$$+ \big(\frac{9}{\lambda_C} + 2L_3^2\big)\frac{\eta_\theta^2}{\eta_\omega}\Big[6L_1^2\mathbb{E}\|\omega_t^{(m)} - \omega^*(\theta_t^{(m)})\|^2 + 9L_1^2\mathbb{E}\|\widetilde{\omega}^{(m)} - \omega^*(\widetilde{\theta}^{(m)})\|^2$$

$$+ 9L_2^2\mathbb{E}\|\theta_t^{(m)} - \widetilde{\theta}^{(m)}\|^2 + \frac{1}{M}\cdot 9C_1 + 9\mathbb{E}\|\nabla J(\theta_t^{(m)})\|^2\Big]$$

$$+ 2\eta_\omega^2\Big[6L_4^2\mathbb{E}\|\omega_t^{(m)} - \omega^*(\theta_t^{(m)})\|^2 + 9L_4^2\mathbb{E}\|\widetilde{\omega}^{(m)} - \omega^*(\widetilde{\theta}^{(m)})\|^2 + 6L_5^2\mathbb{E}\|\theta_t^{(m)} - \widetilde{\theta}^{(m)}\|^2 + \frac{6C_2}{M}\Big].$$

Taking expectation on both sides of the above inequality and simplifying, we obtain that

$$\mathbb{E}\|\omega_{t+1}^{(m)} - \omega^*(\theta_{t+1}^{(m)})\|^2$$

$$\leq \Big(1 - \lambda_C\eta_\omega + \big(\frac{9}{\lambda_C} + 2L_3^2\big)6L_1^2\frac{\eta_\theta^2}{\eta_\omega} + 12L_4^2\eta_\omega^2\Big)\mathbb{E}\|\omega_t^{(m)} - \omega^*(\theta_t^{(m)})\|^2$$

$$+ \Big[\big(\frac{9}{\lambda_C} + 2L_3^2\big)9L_1^2\frac{\eta_\theta^2}{\eta_\omega} + 18L_4^2\eta_\omega^2\Big]\mathbb{E}\|\widetilde{\omega}^{(m)} - \omega^*(\widetilde{\theta}^{(m)})\|^2$$

$$+ \big(\frac{9}{\lambda_C} + 2L_3^2\big)\frac{\eta_\theta^2}{\eta_\omega}\cdot\frac{9C_1}{M} + \eta_\omega^2\cdot\frac{12C_2}{M} + \big(\frac{9}{\lambda_C} + 2L_3^2\big)\frac{\eta_\theta^2}{\eta_\omega}9\mathbb{E}\|\nabla J(\theta_t^{(m)})\|^2$$

$$+ \Big[12L_5^2\eta_\omega^2 + \big(\frac{9}{\lambda_C} + 2L_3^2\big)9L_2^2\frac{\eta_\theta^2}{\eta_\omega}\Big]\mathbb{E}\|\theta_t^{(m)} - \widetilde{\theta}^{(m)}\|^2$$

$$- 2\eta_\omega\mathbb{E}\varkappa_t^{(m)} - 2\eta_\omega\mathbb{E}\varsigma_t^{(m)}, \tag{22}$$

where
$$\varsigma_t^{(m)} := \langle \omega_t^{(m)} - \omega^*(\theta_t^{(m)}), \big(\phi_t^{(m)}(\phi_t^{(m)})^\top - C\big)\big(\omega_t^{(m)} - \omega^*(\theta_t^{(m)})\big)\rangle,$$

and
$$\varkappa_t^{(m)} := \langle \omega_t^{(m)} - \omega^*(\theta_t^{(m)}), \big[(\phi_t^{(m)})^\top \omega^*(\theta_t^{(m)}) - \delta_{t+1}^{(m)}(\theta_t^{(m)})\big]\phi_t^{(m)}\rangle.$$

Applying Lemma D.5, and Lemma D.6 to (22), we obtain that

$$\mathbb{E}\|\omega_{t+1}^{(m)} - \omega^*(\theta_{t+1}^{(m)})\|^2 \le \Big(1 - \frac{1}{2}\lambda_C\eta_\omega + \big(\frac{9}{\lambda_C} + 2L_3^2\big)6L_1^2\frac{\eta_\theta^2}{\eta_\omega} + 12L_4^2\eta_\omega^2\Big)\mathbb{E}\|\omega_t^{(m)} - \omega^*(\theta_t^{(m)})\|^2$$
$$+ \Big[\big(\frac{9}{\lambda_C} + 2L_3^2\big)9L_1^2\frac{\eta_\theta^2}{\eta_\omega} + 18L_4^2\eta_\omega^2\Big]\mathbb{E}\|\widetilde{\omega}^{(m)} - \omega^*(\widetilde{\theta}^{(m)})\|^2$$
$$+ \big(\frac{9}{\lambda_C} + 2L_3^2\big)\frac{\eta_\theta^2}{\eta_\omega}\cdot\frac{9C_1}{M} + \eta_\omega^2\cdot\frac{12C_2}{M} + \big(\frac{9}{\lambda_C} + 2L_3^2\big)\frac{\eta_\theta^2}{\eta_\omega}9\mathbb{E}\|\nabla J(\theta_t^{(m)})\|^2$$
$$+ \Big[12L_5^2\eta_\omega^2 + \big(\frac{9}{\lambda_C} + 2L_3^2\big)9L_2^2\frac{\eta_\theta^2}{\eta_\omega}\Big]\mathbb{E}\|\theta_t^{(m)} - \widetilde{\theta}^{(m)}\|^2$$
$$+ 2\eta_\omega(C_3 + C_4)\frac{1}{M}. \tag{23}$$

Telescoping the above inequality over one epoch, we obtain that

$$\Big(\frac{1}{2}\lambda_C\eta_\omega - \big(\frac{9}{\lambda_C} + 2L_3^2\big)6L_1^2\frac{\eta_\theta^2}{\eta_\omega} - 12L_4^2\eta_\omega^2\Big)\sum_{t=0}^{M-1}\mathbb{E}\|\omega_t^{(m)} - \omega^*(\theta_t^{(m)})\|^2$$
$$\le \mathbb{E}\|\omega_M^{(m)} - \omega^*(\theta_M^{(m)})\|^2$$
$$+ M\Big[\big(\frac{9}{\lambda_C} + 2L_3^2\big)9L_1^2\frac{\eta_\theta^2}{\eta_\omega} + 18L_4^2\eta_\omega^2\Big]\mathbb{E}\|\widetilde{\omega}^{(m)} - \omega^*(\widetilde{\theta}^{(m)})\|^2$$
$$+ \big(\frac{9}{\lambda_C} + 2L_3^2\big)\frac{\eta_\theta^2}{\eta_\omega}\cdot 9C_1 + \eta_\omega^2\cdot 12C_2 + \big(\frac{9}{\lambda_C} + 2L_3^2\big)\frac{\eta_\theta^2}{\eta_\omega}9\sum_{t=0}^{M-1}\mathbb{E}\|\nabla J(\theta_t^{(m)})\|^2$$
$$+ \Big[12L_5^2\eta_\omega^2 + \big(\frac{9}{\lambda_C} + 2L_3^2\big)9L_2^2\frac{\eta_\theta^2}{\eta_\omega}\Big]\sum_{t=0}^{M-1}\mathbb{E}\|\theta_t^{(m)} - \widetilde{\theta}^{(m)}\|^2$$
$$+ 2\eta_\omega(C_3 + C_4).$$

Choosing an appropriate $(\eta_\theta, \eta_\omega)$ such that

$$\frac{1}{2}\lambda_C\eta_\omega - \big(\frac{9}{\lambda_C} + 2L_3^2\big)6L_1^2\frac{\eta_\theta^2}{\eta_\omega} - 12L_4^2\eta_\omega^2 \ge \frac{1}{4}\lambda_C\eta_\omega, \tag{24}$$

and we finally obtain that

$$\sum_{t=0}^{M-1}\mathbb{E}\|\omega_t^{(m)} - \omega^*(\theta_t^{(m)})\|^2$$
$$\le \frac{4}{\lambda_C}\Big[\frac{1}{\eta_\omega} + M\Big[\big(\frac{9}{\lambda_C} + 2L_3^2\big)9L_1^2\frac{\eta_\theta^2}{\eta_\omega^2} + 18L_4^2\eta_\omega\Big]\Big]\mathbb{E}\|\widetilde{\omega}^{(m)} - \omega^*(\widetilde{\theta}^{(m)})\|^2$$
$$+ \frac{4}{\lambda_C}\big(\frac{9}{\lambda_C} + 2L_3^2\big)\frac{\eta_\theta^2}{\eta_\omega^2}\cdot 9C_1 + \frac{4}{\lambda_C}\eta_\omega\cdot 12C_2 + \big(\frac{9}{\lambda_C} + 2L_3^2\big)\frac{\eta_\theta^2}{\eta_\omega^2}\frac{36}{\lambda_C}\sum_{t=0}^{M-1}\mathbb{E}\|\nabla J(\theta_t^{(m)})\|^2$$
$$+ \frac{4}{\lambda_C}\Big[12L_5^2\eta_\omega + \big(\frac{9}{\lambda_C} + 2L_3^2\big)9L_2^2\frac{\eta_\theta^2}{\eta_\omega^2}\Big]\sum_{t=0}^{M-1}\mathbb{E}\|\theta_t^{(m)} - \widetilde{\theta}^{(m)}\|^2$$
$$+ \frac{8}{\lambda_C}(C_3 + C_4).$$

$\square$

**Lemma D.8.** *Under the same assumptions as those of Theorem 4.5, the tracking error can be bounded as*

$$\mathbb{E}\|\widetilde{\omega}^{(m)} - \omega^*(\widetilde{\theta}^{(m)})\|^2 \leq (1 - \frac{1}{2}\lambda_C\eta_\omega)^{mM}\mathbb{E}\|\widetilde{\omega}^{(0)} - \omega^*(\widetilde{\theta}^{(0)})\|^2$$
$$+ \frac{4}{\lambda_C}(C_3 + C_4)\frac{1}{M} + \frac{4}{\lambda_C}H^2\eta_\omega + \frac{2}{\lambda_C}(2L_3^2G^2 + \frac{9}{\lambda_C}G^2)\frac{\eta_\theta^2}{\eta_\omega^2}.$$

*Proof.* Recall the one-step update at $\omega_{t+1}^{(m)}$:

$$\omega_{t+1}^{(m)} = \Pi_R(\omega_t^{(m)} - \eta_\omega h_t^{(m)}).$$

Then, we obtain the following upper bound of the tracking error $\|\omega_t^{(m)} - \omega^*(\theta_t^{(m)})\|^2$.

$$\|\omega_{t+1}^{(m)} - \omega^*(\theta_{t+1}^{(m)})\|^2 \leq \|\omega_t^{(m)} - \omega^*(\theta_t^{(m)}) - \eta_\omega h_t^{(m)} + \omega^*(\theta_t^{(m)}) - \omega^*(\theta_{t+1}^{(m)})\|^2$$
$$\leq \|\omega_t^{(m)} - \omega^*(\theta_t^{(m)})\|^2 - 2\eta_\omega\langle\omega_t^{(m)} - \omega^*(\theta_t^{(m)}), h_t^{(m)}\rangle$$
$$+ 2\langle\omega_t^{(m)} - \omega^*(\theta_t^{(m)}), \omega^*(\theta_t^{(m)}) - \omega^*(\theta_{t+1}^{(m)})\rangle$$
$$+ 2\eta_\omega^2 H^2 + 2\|\omega^*(\theta_t^{(m)}) - \omega^*(\theta_{t+1}^{(m)})\|^2.$$

Then above inequality can be further bounded as

$$\|\omega_{t+1}^{(m)} - \omega^*(\theta_{t+1}^{(m)})\|^2 \leq \|\omega_t^{(m)} - \omega^*(\theta_t^{(m)})\|^2 - 2\eta_\omega\langle\omega_t^{(m)} - \omega^*(\theta_t^{(m)}), h_t^{(m)}\rangle$$
$$+ \lambda_C\eta_\omega\|\omega_t^{(m)} - \omega^*(\theta_t^{(m)})\|^2$$
$$+ (\frac{9}{\lambda_C} + 2L_3^2)\frac{\eta_\theta^2}{\eta_\omega}G^2$$
$$+ 2\eta_\omega^2 H^2.$$

Taking conditional expectation on both sides of the above inequality, we obtain that

$$\mathbb{E}_{m,0}\|\omega_{t+1}^{(m)} - \omega^*(\theta_{t+1}^{(m)})\|^2$$
$$\leq \mathbb{E}_{m,0}\|\omega_t^{(m)} - \omega^*(\theta_t^{(m)})\|^2 - 2\eta_\omega\mathbb{E}_{m,0}\langle\omega_t^{(m)} - \omega^*(\theta_t^{(m)}), H_t^{(m)}(\theta_t^{(m)}, \omega_t^{(m)})\rangle$$
$$- 2\eta_\omega\mathbb{E}_{m,0}\langle\omega_t^{(m)} - \omega^*(\theta_t^{(m)}), -H_t^{(m)}(\widetilde{\theta}^{(m)}, \widetilde{\omega}^{(m)}) + \widetilde{H}^{(m)}\rangle$$
$$+ \lambda_C\eta_\omega\|\omega_t^{(m)} - \omega^*(\theta_t^{(m)})\|^2 + \frac{9\eta_\theta^2}{\lambda_C\eta_\omega}G^2$$
$$+ 2H^2\eta_\omega^2 + 2L_3^2G^2\eta_\theta^2$$
$$= \mathbb{E}_{m,0}\|\omega_t^{(m)} - \omega^*(\theta_t^{(m)})\|^2 - 2\eta_\omega\mathbb{E}_{m,0}\langle\omega_t^{(m)} - \omega^*(\theta_t^{(m)}), H_t^{(m)}(\theta_t^{(m)}, \omega_t^{(m)})\rangle$$
$$+ \lambda_C\eta_\omega\|\omega_t^{(m)} - \omega^*(\theta_t^{(m)})\|^2$$
$$+ 2H^2\eta_\omega^2 + (2L_3^2G^2 + \frac{9}{\lambda_C}G^2)\frac{\eta_\theta^2}{\eta_\omega} \tag{25}$$

To further bound the inequality above, we first consider the following explicit form of the pseudo-gradient term:

$$H_t^{(m)}(\theta, \omega) = [(\phi_t^{(m)})^\top\omega - \delta_{t+1}^{(m)}(\theta)]\phi_t^{(m)}$$
$$= \phi_t^{(m)}(\phi_t^{(m)})^\top(\omega - \omega^*(\theta)) + [(\phi_t^{(m)})^\top\omega^*(\theta_t^{(m)}) - \delta_{t+1}^{(m)}(\theta)]\phi_t^{(m)}$$
$$= (\phi_t^{(m)}(\phi_t^{(m)})^\top - C)(\omega - \omega^*(\theta)) + C(\omega - \omega^*(\theta))$$
$$+ [(\phi_t^{(m)})^\top\omega^*(\theta) - \delta_{t+1}^{(m)}(\theta)]\phi_t^{(m)}. \tag{26}$$

By Assumption 4.3, we have

$$-2\eta_\omega\mathbb{E}_{m,0}\langle\omega_t^{(m)} - \omega^*(\theta_t^{(m)}), C(\omega_t^{(m)} - \omega^*(\theta_t^{(m)}))\rangle \leq -2\eta\lambda_C\|\omega_t^{(m)} - \omega^*(\theta_t^{(m)})\|^2. \tag{27}$$

Substituting eq. (27) and eq. (26) into eq. (25) yields that

$$\mathbb{E}\|\omega_{t+1}^{(m)} - \omega^*(\theta_{t+1}^{(m)})\|^2 \leq (1 - \lambda_C \eta_\omega)\mathbb{E}\|\omega_t^{(m)} - \omega^*(\theta_t^{(m)})\|^2 - 2\eta_\omega \mathbb{E}\varkappa_t^{(m)} - 2\eta_\omega \mathbb{E}\varsigma_t^{(m)}$$
$$+ 2H^2\eta_\omega^2 + \left(2L_3^2 G^2 + \frac{9}{\lambda_C}G^2\right)\frac{\eta_\theta^2}{\eta_\omega}, \tag{28}$$

where

$$\varsigma_t^{(m)} := \langle \omega_t^{(m)} - \omega^*(\theta_t^{(m)}), \left(\phi_t^{(m)}(\phi_t^{(m)})^\top - C\right)\left(\omega_t^{(m)} - \omega^*(\theta_t^{(m)})\right)\rangle,$$

and

$$\varkappa_t^{(m)} := \langle \omega_t^{(m)} - \omega^*(\theta_t^{(m)}), \left[(\phi_t^{(m)})^\top \omega^*(\theta_t^{(m)}) - \delta_{t+1}^{(m)}(\theta_t^{(m)})\right]\phi_t^{(m)}\rangle.$$

Applying Lemma D.5, and Lemma D.6 to the above inequality, we obtain that

$$\mathbb{E}\|\omega_{t+1}^{(m)} - \omega^*(\theta_{t+1}^{(m)})\|^2 \leq (1 - \frac{1}{2}\lambda_C \eta_\omega)\mathbb{E}\|\omega_t^{(m)} - \omega^*(\theta_t^{(m)})\|^2$$
$$+ 2\eta_\omega(C_3 + C_4)\frac{1}{M}$$
$$+ 2H^2\eta_\omega^2 + \left(2L_3^2 G^2 + \frac{9}{\lambda_C}G^2\right)\frac{\eta_\theta^2}{\eta_\omega}.$$

Telescoping the above inequality over one epoch, we obtain that

$$\mathbb{E}\|\omega_M^{(m)} - \omega^*(\theta_M^{(m)})\|^2 \leq (1 - \frac{1}{2}\lambda_C \eta_\omega)^M \mathbb{E}\|\omega_0^{(m)} - \omega^*(\theta_0^{(m)})\|^2$$
$$+ 2\eta_\omega(C_3 + C_4)\frac{1}{M} \cdot \frac{1 - (1 - \frac{1}{2}\lambda_C \eta_\omega)^M}{\frac{1}{2}\lambda_C \eta_\omega}$$
$$+ 2H^2\eta_\omega^2 \cdot \frac{1 - (1 - \frac{1}{2}\lambda_C \eta_\omega)^M}{\frac{1}{2}\lambda_C \eta_\omega} + \left(2L_3^2 G^2 + \frac{9}{\lambda_C}G^2\right)\frac{\eta_\theta^2}{\eta_\omega} \cdot \frac{1 - (1 - \frac{1}{2}\lambda_C \eta_\omega)^M}{\frac{1}{2}\lambda_C \eta_\omega}.$$

By definition, $\widetilde{\omega}^{(m)} = \omega_M^{(m)}$ and $\widetilde{\theta}^{(m)} = \theta_M^{(m)}$, and the initial parameter for the current inner loop is chosen as the reference parameter, $\omega_0^{(m)} = \widetilde{\omega}^{(m)}$ and $\theta_0^{(m)} = \widetilde{\theta}^{(m)}$. Then we have

$$\mathbb{E}\|\widetilde{\omega}^{(m)} - \omega^*(\widetilde{\theta}^{(m)})\|^2 \leq (1 - \frac{1}{2}\lambda_C \eta_\omega)^M \mathbb{E}\|\widetilde{\omega}^{(m)} - \omega^*(\widetilde{\theta}^{(m)})\|^2$$
$$+ 2\eta_\omega(C_3 + C_4)\frac{1}{M} \cdot \frac{1 - (1 - \frac{1}{2}\lambda_C \eta_\omega)^M}{\frac{1}{2}\lambda_C \eta_\omega}$$
$$+ 2H^2\eta_\omega^2 \cdot \frac{1 - (1 - \frac{1}{2}\lambda_C \eta_\omega)^M}{\frac{1}{2}\lambda_C \eta_\omega} + \left(2L_3^2 G^2 + \frac{9}{\lambda_C}G^2\right)\frac{\eta_\theta^2}{\eta_\omega} \cdot \frac{1 - (1 - \frac{1}{2}\lambda_C \eta_\omega)^M}{\frac{1}{2}\lambda_C \eta_\omega}.$$

Then, we unroll the inequality above and yield that

$$\mathbb{E}\|\widetilde{\omega}^{(m)} - \omega^*(\widetilde{\theta}^{(m)})\|^2 \leq (1 - \frac{1}{2}\lambda_C \eta_\omega)^{mM} \mathbb{E}\|\widetilde{\omega}^{(0)} - \omega^*(\widetilde{\theta}^{(0)})\|^2$$
$$+ \frac{4}{\lambda_C}(C_3 + C_4)\frac{1}{M} + \frac{4}{\lambda_C}H^2\eta_\omega + \frac{2}{\lambda_C}\left(2L_3^2 G^2 + \frac{9}{\lambda_C}G^2\right)\frac{\eta_\theta^2}{\eta_\omega^2}.$$

$\square$

# E    OTHER SUPPORTING LEMMAS

**Constant Bounds:**

**Lemma E.1.** *Within the set $\{\theta : \|\theta\| \leq R\}$, there exists a constant $C_{\nabla J}$ such that*

$$\sup_\theta \|\nabla J(\theta)\| \leq C_{\nabla J}. \tag{29}$$

*Proof.* By Lemma E.9, $\nabla J(\theta)$ is smooth. Hence, by the compactness of $\{\theta : \|\theta\| \leq R\}$, we conclude that $\|\nabla J(\theta)\|$ is bounded by a certain constant $C_{\nabla J}$.    $\square$

**Lemma E.2.** *Let $G := r_{\max} + (1 + \gamma)R + \gamma(|\mathcal{A}|Rk_1 + 1)R$ be a constant unrelated to $m$ and $t$. Then $\|G_t^{(m)}\| \leq G$ for all $m$ and $t$.*

*Proof.* By its definition, we obtain that

$$
\begin{aligned}
\|G_t^{(m)}\| &= \|\big( -\delta_{t+1}(\theta_t)\phi_t + \gamma(\omega_t^\top \phi_t)\widehat{\phi}_{t+1}(\theta_t)\big)\| \\
&\leq \|\big( -\delta_{t+1}(\theta_t)\|\|\phi_t\| + \gamma\|(\omega_t^\top \phi_t)\|\|\widehat{\phi}_{t+1}(\theta_t))\| \\
&\leq r_{\max} + (1 + \gamma)R + \gamma(|\mathcal{A}|Rk_1 + 1)R.
\end{aligned}
$$

$\square$

**Lemma E.3.** *Let $H = (2 + \gamma)R + r_{\max}$ be a constant unrelated to $m$ and $t$. Then $\|H_t^{(m)}\| \leq H$ for all $m$ and $t$.*

*Proof.* The result follows from the definition:

$$
\begin{aligned}
\|H_t^{(m)}\| &= \|\big[\phi_t^T \omega_t - \delta_{t+1}(\theta_t)\big]\phi_t\| \\
&\leq (2 + \gamma)R + r_{\max}.
\end{aligned}
$$

$\square$

**Lipschitz Continuity:**

**Lemma E.4.** *The mapping $\omega \mapsto G_{t+1}^{(m)}(\theta, \omega)$ is $L_1$-Lipschitz in $\omega$ for all $\theta$.*

*Proof.* See Lemma 3 of Wang & Zou (2020). $\square$

**Lemma E.5.** *The mapping $\theta \mapsto G_{t+1}^{(m)}(\theta, \omega^*(\theta))$ is $L_2$-Lipschitz in $\theta$.*

*Proof.* See Lemma 3 of Wang & Zou (2020). $\square$

**Lemma E.6.** *The mapping $\omega^*(\cdot)$ is $L_3$-Lipschitz.*

*Proof.* See eq.(56) of Wang & Zou (2020). $\square$

**Lemma E.7.** *The mapping $\omega \mapsto H_{t+1}^{(m)}(\theta, \omega)$ is $L_4$-Lipschitz.*

*Proof.* It follows that

$$
\begin{aligned}
\|H_{t+1}^{(m)}(\theta, \omega_2) - H_{t+1}^{(m)}(\theta, \omega_2)\| &= \|\big(\delta_{t+1}^{(m)}(\theta) - [\phi_t^{(m)}]^T \omega_1\big)\phi_t^{(m)} - \big(\delta_{t+1}^{(m)}(\theta) - [\phi_t^{(m)}]^T \omega_2\big)\phi_t^{(m)}\| \\
&\leq \|\phi_t^{(m)}\|^2 \|\omega_1 - \omega_2\| \\
&\leq \|\omega_1 - \omega_2\|.
\end{aligned}
$$

Hence, $L_4 = 1$. $\square$

**Lemma E.8.** *The mapping $\theta \mapsto H_{t+1}^{(m)}(\theta, \omega^*(\theta))$ is $L_5$-Lipschitz.*

*Proof.* By definition, we have

$$
\begin{aligned}
&\|H_{t+1}^{(m)}(\theta_1, \omega^*(\theta_1)) - H_{t+1}^{(m)}(\theta_2, \omega^*(\theta_2))\| \\
&= \|\big(\delta_{t+1}^{(m)}(\theta_1) - [\phi_t^{(m)}]^T \omega^*(\theta_1)\big)\phi_t^{(m)} - \big(\delta_{t+1}^{(m)}(\theta_2) - [\phi_t^{(m)}]^T \omega^*(\theta_2)\big)\phi_t^{(m)}\| \\
&\leq \|\delta_{t+1}^{(m)}(\theta_1) - \delta_{t+1}^{(m)}(\theta_2)\| + \|\omega^*(\theta_1) - \omega^*(\theta_2)\| \\
&\leq \big((\gamma|\mathcal{A}|k_1 R + 1) + 1 + L_3\big)\|\theta_1 - \theta_2\|.
\end{aligned}
$$

Hence, $L_5 = (\gamma|\mathcal{A}|k_1 R + 1) + 1 + L_3$. $\square$

**Bounding Lyapunov function:**

**Lemma E.9** ($L$-smoothness of $J$). *For any $\theta_1$ and $\theta_2$, it holds that*

$$|J(\theta_1) - J(\theta_2) - \langle \nabla J(\theta_2), \theta_1 - \theta_2 \rangle| \leq \frac{L}{2}\|\theta_1 - \theta_2\|^2.$$

*Proof.* See Lemma 2 of Wang & Zou (2020). $\qquad\square$

**Lemma E.10.** *It holds that*

$$\|\theta_{t+1}^{(m)} - \widetilde{\theta}^{(m)}\|^2 = \eta_\theta^2\|g_t^{(m)}\|^2 + \|\theta_t^{(m)} - \widetilde{\theta}^{(m)}\|^2 - 2\eta_\theta\zeta_t^{(m)}$$
$$+ \eta_\theta\Big[\frac{1}{\beta_t}\|\nabla J(\theta_t^{(m)})\|^2 + \beta_t\|\theta_t^{(m)} - \widetilde{\theta}^{(m)}\|^2\Big], \qquad (30)$$

*where $\zeta_t^{(m)} := \langle g_t^{(m)} - \nabla J(\theta_t^{(m)}), \theta_t^{(m)} - \widetilde{\theta}^{(m)} \rangle$.*

*Proof.* Note that

$$\|\theta_{t+1}^{(m)} - \widetilde{\theta}^{(m)}\|^2 = \|\theta_{t+1}^{(m)} - \theta_t^{(m)}\|^2 + \|\theta_t^{(m)} - \widetilde{\theta}^{(m)}\|^2 + 2\langle\theta_{t+1}^{(m)} - \theta_t^{(m)}, \theta_t^{(m)} - \widetilde{\theta}^{(m)}\rangle$$
$$= \eta_\theta^2\|g_t^{(m)}\|^2 + \|\theta_t^{(m)} - \widetilde{\theta}^{(m)}\|^2 - 2\eta_\theta\langle g_t^{(m)}, \theta_t^{(m)} - \widetilde{\theta}^{(m)}\rangle$$
$$= \eta_\theta^2\|g_t^{(m)}\|^2 + \|\theta_t^{(m)} - \widetilde{\theta}^{(m)}\|^2 - 2\eta_\theta\langle g_t^{(m)} - \nabla J(\theta_t^{(m)}), \theta_t^{(m)} - \widetilde{\theta}^{(m)}\rangle$$
$$- 2\eta_\theta\langle\nabla J(\theta_t^{(m)}), \theta_t^{(m)} - \widetilde{\theta}^{(m)}\rangle$$
$$= \eta_\theta^2\|g_t^{(m)}\|^2 + \|\theta_t^{(m)} - \widetilde{\theta}^{(m)}\|^2 - 2\eta_\theta\langle g_t^{(m)} - \nabla J(\theta_t^{(m)}), \theta_t^{(m)} - \widetilde{\theta}^{(m)}\rangle$$
$$+ \eta_\theta\Big[\frac{1}{\beta_t}\|\nabla J(\theta_t^{(m)})\|^2 + \beta_t\|\theta_t^{(m)} - \widetilde{\theta}^{(m)}\|^2\Big].$$

$$\square$$

## F  DETAILS OF EXPERIMENTS

**Garnet problem:** The Garnet problem Archibald et al. (1995) is specified as $\mathcal{G}(n_\mathcal{S}, n_\mathcal{A}, b, d)$, where $n_\mathcal{S}$ and $n_\mathcal{A}$ denote the cardinality of the state and action spaces, respectively, $b$ is referred to as the branching factor–the number of states that have strictly positive probability to be visited after an action is taken, and $d$ denotes the dimension of the features. In our experiment, we set $n_\mathcal{S} = 5$, $n_\mathcal{A} = 3, b = 2, d = 4$ and generate the features $\Phi \in \mathbb{R}^{n_\mathcal{S} \times d}$ via the uniform distribution on $[0, 1]$. We then normalize its rows to have unit norm. Then, we randomly generate a state-action transition kernel $\mathbf{P} \in \mathbb{R}^{n_\mathcal{S} \times n_\mathcal{A} \times n_\mathcal{S}}$ via the uniform distribution on $[0, 1]$ (with proper normalization). We set the behavior policy as the uniform policy, i.e., $\pi_b(a|s) = n_\mathcal{A}^{-1}$ for any $s$ and $a$. The discount factor is set to be $\gamma = 0.95$. As the transition kernel and the features are known, we compute $\|\nabla J(\theta)\|^2$ to evaluate the performance of all the algorithms. We set the default learning rates as $\eta_\theta = 0.02$ and $\eta_\omega = 0.01$ for both VR-Greedy-GQ and Greedy-GQ algorithm. For VR-Greedy-GQ, we set the default batch size as $M = 3000$.

**Frozen Lake:** We generate a Gaussian feature matrix with dimension 8 to linearly approximate the value function and we aim to evaluate a target policy based on a behavior policy. The target policy is generated via the uniform distribution on $[0, 1]$ with proper normalization and the behavior policy is the uniform policy. We set the learning rates as $\eta_\theta = 0.2$ and $\eta_\omega = 0.1$ for both algorithms and set the batch size as $M = 3000$ for the VR-Greedy-GQ. We run 200k iterations for each of the 10 trajectories.

**Estimated maximum Reward:** In the experiments, we compute the maximum reward as follows: When the policy parameter $\theta_t$ is updated to $\theta_{t+1}$, we estimate the corresponding reward by sampling a Markov decision process $\{s_1, a_1, s_2, a_2, \ldots, s_N, a_N, s_{N+1}\}$ using $\pi_\theta$. Then we estimate the expected reward using

$$\hat{r}_t = \frac{1}{N}\sum_{i=1}^N r(s_i, a_i, s_{i+1}).$$

Under the ergodicity assumption, this average reward will tend to the expected reward with respected the stationary distribution induced by $\pi_\theta$ (Wu et al. (2020)). Then the maximum reward is defined as the maximum estimated expected reward along the training trajectory; that is,

$$\text{Maximum Reward} = \max_t \hat{r}_t.$$

In the experiments, we set $N = 100$ when estimating the expected reward.

