# OpenReview forum: "Greedy-GQ with Variance Reduction: Finite-time Analysis and Improved Complexity"
_ICLR.cc/2021/Conference — ICLR 2021 Poster_

### Official Review · AnonReviewer1 · 2020-10-27
**A variance-reduced Greedy GQ variant along with its sample complexity analysis in the Markov setting**

**Rating:** 8
**Confidence:** 5

**Review:**

This submission deals with the classical value-based Greedy-GQ algorithm for off-policy optimal control, and develops a two-timescale variance reduction scheme to reduce the stochastic variance of Greedy-GQ thus improving its sample complexity.

Specifically, a variance reduced (VR)-Greedy-GQ variant that applies the SVRG-type variance reduction technique to the two-timescale updates of Greedy-GQ. Assuming linear function approximation and Markovian data samples, it is shown that the VR-Greedy-GQ achieves a sample complexity of O(\epsilon^{-2}), which is order-wise lower than the sample complexity O(\epsilon^{-3}) of the original Greedy-GQ. Convincing experiments are also provided to demonstrate the effectiveness of the proposed variance reduction algorithm.

Overall evaluation: This paper is reasonably well written and presents interesting technical results. The Greedy-GQ algorithm is an important and efficient value-based approach for off-policy control.

Detailed comments: In the existing study, variance reduction techniques have been successfully applied to value-based TD learning algorithms for policy evaluation (e.g., VRTD, VRTDC), but they have not been explored by value-based algorithms for control, especially in the off-policy setting with Markovian samples. This paper fills this important gap. Below please find several related technical comments.

i) The main contribution is to show that VR-Greedy-GQ achieves an improved sample complexity over that of Greedy-GQ. In particular, the authors showed that (as commented in the contribution section), VR-Greedy-GQ induces a small bias error caused by the Markovian sampling and a small variance error of the stochastic updates, both errors are inverse proportional to M — the batch size of the SVRG reference batch update. Hence, a larger M should gives smaller error terms, and this is also suggested by the bounds in Theorem 4.5. However, it is not clear why Corollary 4.6 chooses the special M=\epsilon^{-1} to achieve the desired sample complexity, can the author clarify the trade-off in choosing these hyper-parameters?

ii) A key technique in the finite-time analysis is the introduction of the fine-tuned Lyapunov function R_t^m. In particular, the coefficient c_t is specially chosen so that the quadratic term on theta can be totally absorbed into the Lyapunov function for telescoping. Although this technical development is very interesting, how is it different from the traditional analysis of nonconvex SVRG? For example, see the paper Stochastic Variance Reduction for Nonconvex Optimization by Sashank J. Reddi et.al.

iii) More recent results on finite-time analysis of TD/Q-learning algorithms dealing with Markovian samples should be discussed, as well as how the current analysis differentiates/improves from existing e.g., drift analysis in [Srikant et al, COLT'2019] and multistep Lyapunov analysis in [Wang et al, AISTATS'2020] in terms of accommodating the bias and correlations introduced by the Markovian data samples.

---

> ### Author Response · Authors · 2020-11-18
> **Response**
>
> We thank the reviewer for providing valuable feedback that helps us improve the quality of this paper. Below is a point-to-point response to the questions raised by the reviewer.
>
> Q1: It is not clear why Corollary 4.6 chooses the special $M=\epsilon^{-1}$ to achieve the desired sample complexity, can the author clarify the trade-off in choosing these hyper-parameters?
>
> A: Thanks for raising this question. In the complexity bound of Theorem 4.5, there is the term $(\eta_{\theta} / \eta_w)^2 + \eta_w$, which leads to the optimal learning rate relation $\eta_\theta = O(\eta_w^{3/2})$. Then, to satisfy eq.(17), we must choose $\eta_\theta = M^{-1}$. Substituting these choices in the complexity bound yields the order $O(T^{-1} + M^{-1})$. Hence, to achieve an $\epsilon$ accuracy, we need to choose $M = O(\epsilon^{-1})$.
>
> Q2: How is it different from the traditional analysis of nonconvex SVRG?
>
> A: Thanks for raising this question. We agree that part of our analysis of VR-Greedy-GQ follows the standard analysis logic of the conventional nonconvex SVRG, e.g., exploiting the function smoothness and introducing a Lyapunov potential function. However, the major part of the analysis is substantially different from that of the conventional SVRG in the following perspectives:
>
> (i) In order to perform optimal control in the off-policy setting, VR-Greedy-GQ applies two timescale updates and uses Markovian samples, these specialties make the stochastic updates $G_x(\theta, \omega), H_x(\theta, \omega)$ biased estimators of the full gradient $\nabla J(\theta)$. In comparison, conventional SVRG only involves a single timescale update and uses i.i.d samples, which make the stochastic updates unbiased toward the full gradient.
>
> (ii) Moreover, due to the Markovian samples and two timescale updates, our proof needs to develop tight bounds to control the Markovian noise (caused by Markovian samples) and the tracking error (caused by the additional timescale update). These bounds further motivate us to fine-tune the coefficient $c_t$ involved in the Lyapunov function.
>
> With these technical developments, we are able to establish an improved sample complexity result. In the revision, we have clarified in section 5 that the proof partly follows the standard analysis logic of the conventional SVRG, but requires substantial new technical developments to address the challenges of off-policy control.
>
> Q3: More recent results on finite-time analysis of TD/Q-learning algorithms dealing with Markovian samples should be discussed, e.g., drift analysis in [Srikant et al, COLT'2019] and multistep Lyapunov analysis in [Wang et al, AISTATS'2020].
>
> A: We thank the reviewer for pointing out these related works. We have cited them and discussed them in the related work.

---

### Official Review · AnonReviewer2 · 2020-10-27
**The VR method is not SOTA, please justify**

**Rating:** 3
**Confidence:** 5

**Review:**



Greedy-GQ is an RL algorithm for a control problem that extends on GTD, which is a prediction algorithm. While Greedy-GQ asymptotically converges to a stationary point, it does so with high sample complexity. The authors reduce the variance of Greedy-GQ by incorporating SVRG variance reduction scheme to both the time-scale update of the algorithm. The main contribution of the paper is in showing that the variance reduced Greedy-GQ algorithm achieves a sample complexity that is an order of magnitude less than the vanilla Greedy-GQ.

Major concerns:

1.	My major concern is that the paper uses SVRG for variance reduction, which is not the state-of-the-art method. Since the two seminal methods (SVRG and SAGA) have been proposed, there have been substantial achievements during the past few years along this direction, such as SARAH, SPIDER, and STORM. It is very important they author identify with enough analysis that those three competing methods as mentioned above reach the same time and space complexity (aka, w.r.t the mini-batch size). The reviewer strongly suggests the author to do so.
2.	Assumption 4 (geometric ergodicity) seems too restrictive. The author should give some more detailed justification to justify if it is widely applicable or not.
3.	For the experiments, it would have been nice to include the plots based on the objective function and not just its norms.
4.	Wordings of the experiments in 6.2 is a bit confusing to me. After the target policy is generated via the uniform distribution, is it getting improved as the off-policy control algorithm should, or is it only evaluated against the behavior policy?
5.	in section 1.2 the paper concluded Q-learning and SARSA with function approximation as related work. However, the proposed algorithm doesn’t seem to be related to it.
6.  I noticed that there is a recently published NeurIPS paper (Variance-Reduced Off-Policy TDC Learning: Non-Asymptotic Convergence Analysis), which further reduces the merit of the paper on the perspective of adapting SVR techniques to nonstandard stochastic optimization algorithms such as TDC/Greedy-GQ.


Minor concerns:
1.	Page 6, Corollary 4.6: Shouldn’t one of the \eta_\theta be \eta_\omega?
2.	in section 1.1 paragraph 2 line 7, it is better to use “order of O(M^-1) and O(\eta_\theta M^-1) respectively” instead of “…O(M^-1), O(\eta_\theta M^-1),…”
3.	in section 5 equation (6), it is better to use “… 1/M)” instead of “…M^-1” to keep consistency.
4.	In section 5 step 3, it is better to say lemma D.7 and D.9 is in the Appenix.
5.	The paper mentioned “SPIDER (a.k.a SARAH)”. These two algorithms are very similar, but they are not identical, as pointed out by several papers. For example, referring to the last few paragraphs of pp.3 of the paper “Finite-Sum Smooth Optimization with SARAH” (https://arxiv.org/pdf/1901.07648.pdf).

---

> ### Author Response · Authors · 2020-11-18
> **Response**
>
> We thank the reviewer for providing valuable feedback that helps us improve the quality of this paper. Below is a point-to-point response to the questions raised by the reviewer.
>
> Q1: There have been substantial developments such as SARAH, SPIDER, and STORM. It is very important the author identifies with enough analysis that those three competing methods as mentioned above reach the same time and space complexity.
>
> A: We understand that there are various successful variance reduction techniques in the nonconvex stochastic optimization area, and we believe that SVRG is one of the representative ones. Studying Greedy-GQ with all these different variance reduction techniques is beyond the limit of a conference paper, given that we are dealing with a complex problem with Markovian samples and two timescale updates. Instead, we focus on a representative approach and provide a comprehensive study with an improved state-of-the-art complexity result.
>
> Q2: Assumption 4 (geometric ergodicity) seems too restrictive
>
> A: We want to clarify that this is a standard assumption that is widely adopted in the existing literature, see the finite time analysis of TD (Bhandari et.al 2018), VRTD (Xu et.al 2020), TDC (Xu et.al 2019), Q-learning (Xu & Gu 2019), Greedy-GQ (Zou & Wang 2020), etc. In particular, all uniformly ergodic Markov chains satisfy this assumption. To further elaborate, a Markov chain is called uniformly ergodic if it is irreducible (i.e., possible to reach any state $s_a$ from any other state $s_b$) and aperiodic (Levin & Peres 2017). These basic properties of the environment are commonly seen in RL practice.
>
>
> Q3: For the experiments, it would have been nice to include the plots based on the objective function and not just its norms.
>
> A: We thank the reviewer for the suggestion. In the revision, we provided the objective function vs # gradient computation plot in the Figure 2 (left) and Figure 4 (left) for both Greedy-GQ and VR-Greedy-GQ. It can be seen that our VR-Greedy-GQ achieves a much lower objective function value than Greedy-GQ.
>
> Moreover, as suggested by other reviewers, we also plotted the expected reward vs # of samples plot in the Figure 2 (right) and Figure 4 (right) for Greedy-GQ, VR-Greedy-GQ and policy gradient. The plots demonstrate that our VR-Greedy-GQ can effectively achieve a higher reward than Greedy-GQ and policy gradient.
>
>
> Q4: The wording of the experiments in 6.2 is a bit confusing to me. After the target policy is generated via the uniform distribution, is it getting improved as the off-policy control algorithm should, or is it only evaluated against the behavior policy?
>
> A: Sorry for the confusion. To clarify, the target policy is updated by the policy improvement operator in every iteration. The behavior policy is fixed throughout.
>
> Q5: In Section 1.2 the paper concluded Q-learning and SARSA with function approximation as related work. However, the proposed algorithm doesn’t seem to be related to it.
>
> A: We think the works on Q-learning and SARSA with linear function approximation are related. In fact, Q-learning and SARSA were known to possibly diverge in the off-policy setting under linear function approximation (Baird 1995), and this further motivates the design of the Greedy-GQ algorithm, which uses a two timescale update to address this issue.
>
> Q6: Minor concerns.
>
> A: We thank the reviewer for pointing out these issues. We have fixed them in the revision. In particular, we do agree that SARAH and SPIDER are different algorithms.

---

### Official Review · AnonReviewer4 · 2020-10-28
**A novel algorithm with convergence analysis.**

**Rating:** 6
**Confidence:** 4

**Review:**

This paper proposes a variance reduced Greedy-GQ algorithm and proves that it enjoys a lower convergence rate compared with vanilla Greedy-GQ. The paper is well written and clearly presented. The sample complexity of the proposed algorithm is lower than Greedy-GD by a factor of 1/eps, which could be a great improvement when the target precision eps is small.

In the related work part, the nonasymptotic convergence of Q-learning with function approximation has also been extended to neural network function approximation (A finite-time analysis of q-learning with neural network function approximation. In ICML 2020).

In the algorithm, what specific policy improvement is used? Does the choice affect the proof of the convergence?

One drawback of the result in Theorem 4.5 is that the convergence is only guaranteed for the minimal gradient norm over the whole trajectory. However, in Algorithm 1, the output is defined as the last iterate. In other words, the convergence rate in Theorem 4.5 is not for the proposed algorithm. In this sense, the convergence results may not be as useful as other algorithms that can guarantee the convergence of the last iterate of the average iterate.

In terms of the experiments, it would be nice if the cumulative reward versus running time can be shown, which demonstrates the performance in a clearer way. It would also be interesting to see whether the proposed variance reduced Greedy-GQ method can match the performance of off-policy policy gradient (or actor-critic) methods and variance reduced policy gradient methods in controlling problems.

#########Edits after the rebuttal#########

I have read the response and other reviewers' review/discussion. I will keep a score as 6.

---

> ### Author Response · Authors · 2020-11-18
> **Response**
>
> We thank the reviewer for providing valuable feedback that helps us improve the quality of this paper. Below is a point-to-point response to the questions raised by the reviewer.
>
> Q1: In the related work, the nonasymptotic convergence of Q-learning with function approximation has also been extended to neural network function approximation (A finite-time analysis of q-learning with neural network function approximation. In ICML 2020).
>
> A: We thank the reviewer for pointing out this related work. We have cited this work in the revision.
>
> Q2: In the algorithm, what specific policy improvement is used? Does the choice affect the proof of the convergence?
>
> A: Our proof applies to all smooth policy improvements, examples include softmax, mellowmax, etc. These different choices of policy improvement do not affect our technical proof.  We commented on this after Assumption 4.2.
>
> Q3: One drawback of Theorem 4.5 is that the convergence is only guaranteed for the minimal gradient norm over the trajectory. However, the output of Algorithm 1 is defined as the last iterate.
>
> A: Thanks for pointing out this mismatch, and we would like to provide a simple fix of this problem. Note that on **page 19 of the revision**, we derived a bound on the average gradient norm $\frac{1}{TM} \sum_{m,t} \mathbb{E}||\nabla J(\theta_t^{(m)})||^2$, which immediately leads to the minimum gradient norm bound. We note that this average gradient norm bound can be equivalently expressed as the expectation bound $\mathbb{E}||\nabla J(\theta_{\xi}^{(\zeta)})||^2$, where $\xi$ is a random index sampled from $0,..., M-1$ and $\zeta$ is a random index sampled from $1,..., T$, both uniformly at random. Therefore, we propose to replace the minimum norm bound in Theorem 4.5 by this expectation bound, and correspondingly change the output of the Algorithm 1 to be the iteration that is uniformly sampled among all the past iterations. We have updated the bound in Theorem 4.5 and the Algorithm output in the revision.
>
> We also note that such a uniformly sampling output strategy can be implemented in a way similar to the last iterate output strategy, i.e., before running the algorithm, we first draw the random numbers $\xi, \zeta$ from ${0,..., M-1}$ and ${1,...,T}$, respectively, uniformly at random. Then we run the algorithm for $(\xi-1)M+\zeta$ iterations and output the last iterate.
>
>
> Q4: It would be nice if the cumulative reward versus running time can be shown. It would also be interesting to see whether the proposed method can match the performance of off-policy policy gradient (or actor-critic) methods and variance reduced policy gradient methods.
>
> A: We thank the reviewer for these suggestions. For these two experiments, we have plotted the estimated expected reward v.s. # of samples for Greedy-GQ, VR-Greedy-GQ and policy gradient in the Figure 2 (right) and Figure 4 (right) (see the revision). It can be seen from these figures that VR-Greedy-GQ does achieve a higher expected reward and is the most sample efficient algorithm. This demonstrates the effectiveness of variance reduction of our method.
> Due to limited time, we have not finished comparing with other algorithms suggested by the reviewer, but we will update the results in the revision.

---

> ### Author Response · Authors · 2020-11-23
> **Extra experiments added**
>
> We just added some experiments on the comparison between VR-Greedy-GQ and actor-critic to the paper. Please find the comparison results in Figure 2 (middle) and Figure 4 (middle), where the $y$-axis denotes the expected reward and the $x$-axis denotes the number of gradient computations. In these experiments, our VR-Greedy-GQ consistently outperforms the standard actor-critic algorithm and achieves a higher expected reward.

---

### Official Review · AnonReviewer3 · 2020-10-30
**Interesting paper that addresses variance issues with Greedy-GQ**

**Rating:** 8
**Confidence:** 3

**Review:**

EDIT: After reading the other reviews, responses, and thinking more about the issues raised and resolved, I'm increasing my score to an 8.

---

### Summary:
The paper introduces a variance-reduced version of the Greedy-GQ algorithm for off-policy control, based on SVRG. The paper then analyzes the finite-time convergence rate of the proposed method (VR-Greedy-GQ) under the assumption of Markovian noise and finds an improvement on the order of 1/epsilon to achieve an epsilon-stationary point. The theoretical findings are supported by some experiments.

### Pros:
- Proposed improvement to existing method
- Theoretical support for proposed improvement
- Markovian noise assumption
- Some new techniques in the theoretical analysis
- Well written

### Cons:
- SVRG setup involving inner and outer loop conflicts with the original motivation for Greedy-GQ (online and incremental latent learning)

#### Clarity/Quality:
The paper is well written and not too difficult to follow despite the complicated topics. Sections 4 (Finite-time analysis of VR-Greedy-GQ) and 5 (sketch of the technical proof) were especially well-written and helpful, and the experiments seemed reasonable.

#### Originality:
The paper seems fairly incremental in terms of the proposed method, applying an existing method for variance reduction to an existing algorithm. However, some of the technical tools used in the analysis of the resulting VR-Greedy-GQ algorithm are original to the best of my knowledge.

### Decision:
Despite some concerns listed below, I recommend accepting the paper for publication.

Why use SVRG to reduce variance? The nested loop structure seems at odds with the single Markovian path of experience, and conflicts with the original motivation for Greedy-GQ ("Online, incremental, with memory and per-time-step computation costs that are linear in the number of features" ). If we're giving up online and incremental algorithms, why not use an LSTD-based method like LSPI (Lagoudakis & Parr, 2003) instead?

Why require smooth policies when the original Greedy-GQ paper doesn’t? Is it just for convenience? Would the analysis still apply for a non-smooth policy like the greedy policy?

### Suggestions for improvement:
- In the related work section, it could be helpful to explicitly write how the related work is different from the current work.

### Miscellaneous comments:
- Spelling error in third paragraph of section 1.1: “new tecnical developments”
- In section 2.2 the paper states that “[Greedy-GQ] aims to minimize the Bellman error”, but my understanding is that Greedy-GQ was designed explicitly for the function approximation setting where minimizing MSBE and MSPBE are different. It would be better to write “and in the tabular setting it aims to minimize the Bellman error”.
- Corollary 4.2 sets $\eta_\theta$ twice. Is the second $\eta_\theta$ supposed to be $\eta_\omega$?

### References:
- Wang, Y., & Zou, S. (2020). Finite-sample Analysis of Greedy-GQ with Linear Function Approximation under Markovian Noise. arXiv preprint arXiv:2005.10175.
- Maei, H. R., Szepesvári, C., Bhatnagar, S., & Sutton, R. S. (2010, January). Toward off-policy learning control with function approximation. In ICML.
- Lagoudakis, M. G., & Parr, R. (2003). Least-squares policy iteration. Journal of machine learning research, 4(Dec), 1107-1149.

---

> ### Author Response · Authors · 2020-11-18
> **Response**
>
> We thank the reviewer for providing valuable feedback that helps us improve the quality of this paper. Below is a point-to-point response to the questions raised by the reviewer.
>
> Q1: Why use SVRG to reduce variance? The nested loop structure seems at odds with the single Markovian path of experience, and conflicts with the original motivation for Greedy-GQ
>
> A: Our VR-Greedy-GQ is based on the online-SVRG and can be viewed as an incremental algorithm with regard to the batches of samples used in the outer-loops. To explain, according to Algorithm 1, in every $m$-th outer-loop, we first obtain a new batch $B_m$ of samples from the single MDP trajectory. Then, in the corresponding inner-loops, we exploit this batch of samples to perform SVRG variance reduction. Hence, the algorithm incrementally samples new batches of samples in the outer loops and uses them to perform variance reduction. In this sense, VR-Greedy-GQ can be viewed as a batch-incremental algorithm.
>
> We note that in general, there is a trade-off between incrementalism and variance reduction for SVRG-type algorithms: a larger batch size in the outer-loops enhances the effect of variance reduction, while a smaller batch size makes the algorithm more incremental. We also note that although every outer-loop samples a batch of $B_m = M$ samples, the batch reference updates $\tilde{G}^{(m)}, \tilde{H}^{(m)}$ are reused $M$ times in the inner loops. This is a property of SVRG that makes the memory and computation cost per iteration the same as the incremental SGD.
>
> In the revision, we have added a paragraph to discuss and clarify the online and incremental property of VR-Greedy-GQ (see the last paragraph of Section 3).
>
>
> Q2: Why require smooth policies when the original Greedy-GQ paper doesn’t? Is it just for convenience? Would the analysis still apply for a non-smooth policy like the greedy policy?
>
> A: Thanks for raising this question. In fact, many of the smooth policies, e.g., softmax policy and mellowmax policy, converge to the greedy policy if their temperature parameters tend to $+\infty$. Moreover, smooth policies are preferred in practice over greedy-policy as they usually lead to more stable training. Our analysis leverages the smoothness of the objective function under the smooth policy, which may not hold under the greedy policy. For the analysis under the greedy policy, we may need to develop a subgradient descent-type analysis to address the non-smoothness, and this typically leads to a worse complexity bound.
>
> Q3: In the related work section, it could be helpful to explicitly write how the related work is different from the current work.
>
> A: We thank the reviewer for the suggestion. We have clarified the differences from the existing studies in the related work section.
>
>
> Q4: Miscellaneous comments.
>
> A: We thank the reviewer for pointing out these typos and errors. We have corrected them in the revision. To clarify, in Corollary 4.2, the first equation $\eta_\theta = O(\eta_\omega^{3/2})$ specifies the relation between $\eta_\theta$ and $\eta_\omega$, and the second equation specifies the choice of $\eta_\theta$. We have rewritten this in a clear way.

---

> > ### Comment · AnonReviewer3 · 2020-11-23
> > **Thank you for your response**
> >
> > Thank you for responding to my review and addressing my questions.

---

### Official Review · AnonReviewer5 · 2020-11-06

**Rating:** 5
**Confidence:** 3

**Review:**

This paper combines a widely used variance reduction technique SVRG with the greedy-GQ. It provides a finite-time analysis of the proposed algorithm  in the off-policy and Markovian sampling setting (convergence to the stationary point) and improves the sample complexity from the order $\epsilon^{-3}$ to $\epsilon^{-2}$ comparing with the vanilla greedy GQ. Interestingly, the analysis shows that the biase error caused by the Markovian sampling and the variance error of the stochastic gradient are reduced by the $M$, where M is the batch size of the batch gradient in SVRG. At last, it verifies the theoretical claim by two toy examples.

pros:
1. It combines the variance reduction trick in optimization community with the two time scale analysis in RL.
2. The analysis is on the off-policy control setting, which in general is much harder than the off-policy evaluation setting.
3. The objective function of MSPBE in control setting is non-convex, which increases the difficulty of the proof.


cons:
1. The main contribution of this paper is its theoretical analysis. However the techniques in the proof have already existed in many literatures. It seems that the author just combines them together. For instance, there are many literatures on the convergence analysis of the SVRG in the non-convex setting.  The author claims that the 'fine-tuned' Lyapunov function is novel. However such tools in the non-convex SVRG are widely used. It may be true that we need to chose the c_t carefully to cancel some error term but  the main framework of the proof is the same.

2. I am not sure whether the variance reduction technique is useful in practice. There are some evidences that SVRG does not work well in the training of deep learning problem.  Would faster convergence to the stationary points lead to the better performance (e.g. higher reward )? I do not see anything related to that in the experiment.

3. The author just tests their algorithm on two toy examples. I hope to see more complicated experiments. Maybe the author can try the neural network in the function approximation beyond the linear function approximation.  I know the analysis is just on the linear case, but this experiment would demonstrate the potential applicability of the algorithm.


################after rebuttal

After reading the responds from the author, I keep my score at 5.

---

> ### Author Response · Authors · 2020-11-18
> **Response**
>
> We thank the reviewer for providing valuable feedback that helps us improve the quality of this paper. Below is a point-to-point response to the questions raised by the reviewer.
>
> Q1: Proof techniques already exist. It seems that the author just combines them together.
>
> A: We agree that not every technique used in this paper is new, and part of our analysis logic of VR-Greedy-GQ follows the standard analysis logic of the conventional nonconvex SVRG, i.e., exploiting the function smoothness and introducing a Lyapunov function. However, the majority part of the analysis is substantially different from that of the conventional SVRG in the following perspectives:
>
> (i) In order to perform optimal control in the off-policy setting, VR-Greedy-GQ applies two timescale updates and uses Markovian samples, these specialties make the stochastic updates $G_x(\theta, \omega), H_x(\theta, \omega)$ biased estimators of the full gradient $\nabla J(\theta)$. In comparison, conventional SVRG only involves a single timescale update and uses i.i.d. samples, which make the stochastic updates unbiased toward the full gradient.
>
> (ii) Due to the Markovian samples and two timescale updates, our proof needs to develop tight bounds to control the Markovian noise (caused by Markovian samples) and the tracking error (caused by the additional timescale update). These bounds further motivate us to fine-tune the coefficient $c_t$ involved in the Lyapunov function.
>
> With these technical developments, we are able to establish an improved sample complexity result. In the revision, we have clarified in the beginning of Section 5 and in the contribution section, that the proof partly follows the standard analysis logic of the conventional SVRG, but requires substantial new technical developments to address the challenges of off-policy control, two time-scale updates and Markovian samples.
>
>
> Q2: Would faster convergence to the stationary points lead to a higher reward?
>
> A: We thank the reviewer for the suggestion. For these two experiments, we have plotted the estimated expected reward v.s. # of samples for Greedy-GQ, VR-Greedy-GQ and policy gradient (as suggested by Reviewer 4) in Figures 2 (Right) and Figure 4 (Right) (see the revision). It can be seen from these figures that VR-Greedy-GQ does achieve a much higher expected reward.
>
>
> Q3: More interesting to apply neural network approximation.
>
> A: We thank the reviewer for pointing out this interesting direction. The vanilla Greedy-GQ algorithm derived in Maei et.al. (2010) heavily relies on the linear approximation structure and the mean-square projected Bellman error (MSPBE). It turns out that Neural network approximation is not directly applicable to the vanilla Greedy-GQ and our variance-reduced Greedy-GQ algorithms.
>
> To explain, under linear approximation, the projection operator in MSPBE projects the Bellman error to the linear function space and has a simple closed form. However, under neural network approximation, this projection operator becomes highly non-trivial, i.e., the projection onto the neural network function space does not have a closed form solution and thus cannot be computed in an online and incremental fashion. We note that one approach was proposed in Maei et al. (2009) to address a similar issue of using arbitrary smooth function approximation, under which the projection is onto the tangent plane of the function approximation at the current estimate of $\theta$, and therefore, the projection operator depends on the time-varying parameter $\theta$. We note that this approach in Maei et al. (2009) was proposed for policy evaluation, not for optimal control. It remains to be understood whether such an approach can be extended to the optimal control problem and help design a Greedy-GQ-type algorithm. We think this is an interesting topic that is worth further exploration.

---

### Decision · Program_Chairs · 2021-01-07
**Final Decision**

**Decision:**

Accept (Poster)

**Comment:**

**Overview** This paper provides a way to combine SVRG and greedy-GQ to improve the algorithm performance. In particular, the finite iteration complexity is improved from $\epsilon^{-3}$ to $\epsilon^{-2}$.

**Pros** The paper is well-written. Reviewers believe this is a solid theoretical work on advancing value-based algorithms for off-policy optimal control. It has sufficient theoretical advancement and experiments demonstrations of the methods.

**Cons** Some reviewers are concerned that SVRG is not SOTA. SVRG is not used in practice. The techniques appear to be similar to some existing works.

**Recommendation** The meta-reviewer believes that the paper has solid theoretical contributions. SVRG is a component in the new algorithm to improve the complexity. It does not need to be "useful" or "SOTA". The paper is also well-written. Hence the recommendation is accept.